# Adversarial Robustness of Graph Transformers

**Philipp Foth**[*]                                                                          *p.foth@tum.de*
*School of Computation, Information and Technology*
*Technical University of Munich*

**Lukas Gosch**[*]                                                                          *l.gosch@tum.de*
*School of Computation, Information and Technology & Munich Data Science Institute*
*Technical University of Munich; Munich Center for Machine Learning (MCML)*

**Simon Geisler**[†]                                                                          *s.geisler@tum.de*
*School of Computation, Information and Technology & Munich Data Science Institute*
*Technical University of Munich*

**Leo Schwinn**                                                                          *l.schwinn@tum.de*
*School of Computation, Information and Technology & Munich Data Science Institute*
*Technical University of Munich*

**Stephan Günnemann**                                                                          *s.guennemann@tum.de*
*School of Computation, Information and Technology & Munich Data Science Institute*
*Technical University of Munich; Munich Center for Machine Learning (MCML)*

**Reviewed on OpenReview:** *https://openreview.net/forum?id=4xKOvjxTWL*

## Abstract

Existing studies have shown that Message-Passing Graph Neural Networks (MPNNs) are highly susceptible to adversarial attacks. In contrast, despite the increasing importance of Graph Transformers (GTs), their robustness properties are unexplored. We close this gap and design the first adaptive attacks for GTs. In particular, we provide *general design principles* for strong gradient-based attacks on GTs w.r.t. structure perturbations and instantiate our attack framework for five representative and popular GT architectures. Specifically, we study GTs with specialized attention mechanisms and Positional Encodings (PEs) based on pairwise shortest paths, random walks, and the Laplacian spectrum. We evaluate our attacks on multiple tasks and perturbation models, including structure perturbations for node and graph classification, and node injection for graph classification. Our results reveal that GTs can be *catastrophically fragile* in many cases. Addressing this vulnerability, we show how our adaptive attacks can be effectively used for adversarial training, substantially improving robustness.

## 1 Introduction

Graphs are fundamental data structures with broad applications across various domains. In recent years, Graph Neural Networks (GNNs) have become the go-to method for learning on graph-structured data. Given their growing adoption, numerous studies have explored adversarial attacks on GNNs, revealing their susceptibility to even minor perturbations of the graph structure (Zügner et al., 2018; Zügner & Günnemann, 2019; Zügner & Günnemann, 2020). These studies mainly focus on Message-Passing GNNs (MPNNs), such as Graph Convolutional Networks (GCNs) (Kipf & Welling, 2017). More recently, Graph Transformer (GT) models have emerged as a promising alternative, addressing key limitations of MPNNs,

---

[*]Equal contribution. Correspondence should be addressed to *l.gosch@tum.de*.
[†]Now with Google Research.

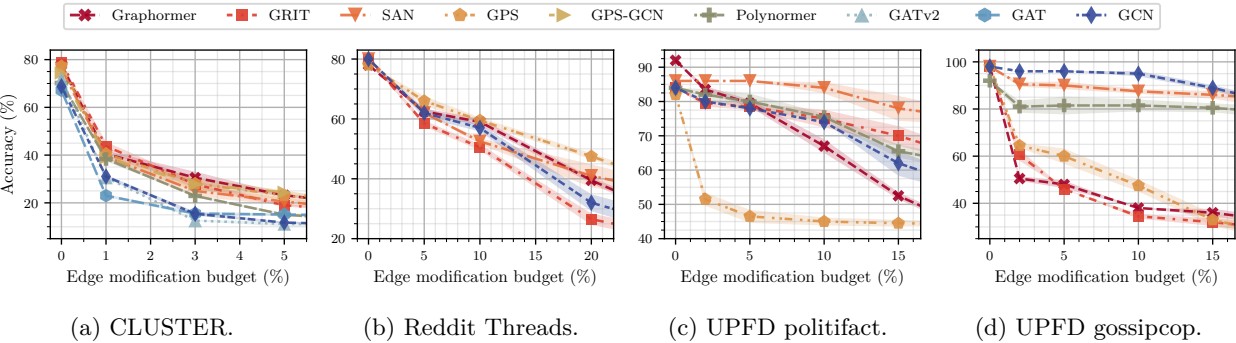

(a) CLUSTER.  (b) Reddit Threads.  (c) UPFD politifact.  (d) UPFD gossipcop.

Figure 1: The adversarial classification accuracy for different GNNs with varying (evasion) attack budgets on four different datasets: CLUSTER - inductive node classification (global structure attack), Reddit Threads - graph classification (structure attack), UPFD politifact and gossipcop - graph classification (node injection attack). The strongest attack (out of 9, see Section 5 for more details) for each budget is shown.

such as *over-smoothing*, *over-squashing*, and limited receptive fields (Müller et al., 2024). Despite their growing popularity, the adversarial robustness of GTs is unexplored and hence, unknown. This gap highlights a crucial limitation in our understanding of GTs and poses a risk in practical applications where robustness is critical. However, to understand their robustness, it is not possible to directly apply state-of-the-art attacks for GNNs, such as PGD (Xu et al., 2019) and PRBCD (Geisler et al., 2021) as GTs employ modified attention mechanisms and Positional Encodings (PEs) that are not differentiable w.r.t. the input. Consequently, the lack of good tools to evaluate the robustness of GTs makes it difficult to understand the robustness properties of different GT models, to determine which models or components are preferable in safety-critical settings, and to apply state-of-the-art defense mechanisms such as adversarial training (Gosch et al., 2023a).

To address this challenge, we establish *general guiding principles* for designing differentiable relaxations to the discrete and non-differentiable components in GTs. In doing so, we provide a general outline on how to design strong gradient-based adaptive attacks for GTs that can adjust to all relevant architectural details. Such adaptive attacks are essential for realistic robustness estimates in the vision domain (Athalye et al., 2018; Carlini & Wagner, 2017; Tramèr et al., 2020) as well as the GNN domain (Mujkanovic et al., 2022). We exemplify our guiding principles by developing specific relaxations for the most widely used GT components including **(a) Shortest Path**, **(b) Random Walk**, and **(c) Spectral** PEs. Using our relaxations, we provide the first analysis of the robustness of GTs by applying adaptive gradient-based attacks to five popular and representative GT architectures: **1) Graphormer** (Ying et al., 2021), **2)** Spectral Attention Network (**SAN**) (Kreuzer et al., 2021), **3)** Graph Inductive bias Transformer (**GRIT**) (Ma et al., 2023), **4)** General, Powerful, Scalable (**GPS**) GT (Rampášek et al., 2022), and **5) Polynormer** (Deng et al., 2024).

Our study reveals that GTs can be catastrophically fragile if evaluated with our adaptive attacks (Fig. 1). For example, with our proposed node injection attacks (NIAs), perturbing 2% of the edges can halve the model's accuracy (Fig. 1c & 1d). Consequently, we use our adaptive attacks to devise an effective adversarial training strategy and show its potential to alleviate the hypersensitivity of GT architectures.

Our *main contributions* are:

**(1)** We formulate general guiding principles to relax non-differentiable GT (Graph Transformer) components. Based on this, we develop the first adaptive gradient-based structure attacks for five representative GT architectures. Our developed relaxations concern the most common building blocks found in GTs and thus, can find application across many different GT models.

**(2)** We conduct the first principled empirical study into the adversarial robustness of GTs and show that they can suffer from catastrophic vulnerabilities to even minor perturbations of the graph's structure, in some cases even worse than traditional message-passing GNNs.

**(3)** We show how to leverage our adaptive attacks for adversarial training strategies that can result in an effective defense countering GTs' vulnerabilities. Thus, we establish that the flexibility of GT models can lead to significantly better robust learning capabilities compared to classic message-passing GNNs.

## 2 Preliminaries

Let $\mathcal{G} = (\mathcal{V}, \mathcal{E})$ be an undirected attributed graph with $n$ nodes $\mathcal{V} = \{v_1, ..., v_n\}$ and $m$ edges. Let $\boldsymbol{x}_i \in \mathbb{R}^d$ be the feature vector of node $v_i$. Then the graph can be defined as $\mathcal{G} = (\boldsymbol{A}, \boldsymbol{X})$ with its symmetric binary adjacency matrix $\boldsymbol{A} \in \{0,1\}^{n \times n}$ and node feature matrix $\boldsymbol{X} \in \mathbb{R}^{n \times d}$. The diagonal degree matrix $\boldsymbol{D}$ with entries $\boldsymbol{D}_{ii} = \deg(v_i) = \sum_{j=1}^{n} \boldsymbol{A}_{ij}$ and the normalized symmetric graph Laplacian matrix $\boldsymbol{L}_{sym} = \boldsymbol{I} - \boldsymbol{D}^{-1/2} \boldsymbol{A} \boldsymbol{D}^{-1/2}$ can both be derived from $\boldsymbol{A}$. The GNNs considered in this work are functions $f_\theta(\boldsymbol{A}, \boldsymbol{X})$ with model parameters $\theta \in \mathbb{R}^p$. We denote the updated hidden node representations after each GNN layer $l$ as $\boldsymbol{H}^{(l)}$ with initialization $\boldsymbol{H}^{(0)} = \boldsymbol{X}$. For node-level tasks, we assume that each node should be assigned a class $c \in \{1, \ldots, K\}$ and the output node representations are directly utilized for the prediction, while for graph-level tasks, a graph-pooling operation aggregates the node embeddings into a graph embedding before predicting one out of $K$ classes for the whole graph.

### 2.1 Structure Attacks

In this work, we focus on *untargeted* white-box *evasion* attacks, i.e., an attacker with full knowledge of the model and data attempts to change the trained model's prediction to any incorrect class at test time by slightly perturbing the input graph structure. For node-level tasks we focus on *global* attacks that minimize the overall performance metric across all nodes. The attack objective is described by the following optimization problem:

$$\max_{\tilde{\boldsymbol{A}} \text{ s.t. } ||\tilde{\boldsymbol{A}} - \boldsymbol{A}||_0 < \Delta} \mathcal{L}_{atk}(f_\theta(\tilde{\boldsymbol{A}}, \boldsymbol{X})) \tag{1}$$

where $f_\theta$ is the GNN model with fixed parameters $\theta$, $\tilde{\boldsymbol{A}} \in \{0,1\}^{n \times n}$ is the discrete perturbed adjacency matrix in relation to $\boldsymbol{A}$ with the number of edge flips bounded by the budget $\Delta \in \mathbb{N}_0$, and $\mathcal{L}_{atk}$ is a suitable attack loss function. For node classification, we use the *tanh-margin* attack loss proposed in Geisler et al. (2021). For graph classification, we optimize the unnormalized class logits: $\mathcal{L}_{atk} = -l_y + \sum_{c \neq y} l_c$, where $l_c \in \mathbb{R}$ refers to the unnormalized logit of class $c \in \{1, \ldots, K\}$. It is convenient to model the perturbation as a function of the binary matrix indicating the edge flips $\boldsymbol{B} \in \{0,1\}^{n \times n}$:

$$\tilde{\boldsymbol{A}} = \boldsymbol{A} + \delta \boldsymbol{A}, \qquad \delta \boldsymbol{A} = (\boldsymbol{1}_n \boldsymbol{1}_n^\top - 2\boldsymbol{A}) \odot \boldsymbol{B} \tag{2}$$

with element-wise product $\odot$. Often, the combinatorial problem in Eq. 1 can be optimized more efficiently using a continuous relaxation $\boldsymbol{B}' \in [0,1]^{n \times n}$ replacing $\boldsymbol{B}$ in Eq. 2. In this setting, the entry $\boldsymbol{B}'_{ij}$ represents the probability that the edge $(v_i, v_j)$ is flipped. Then, the discrete perturbation matrix $\tilde{\boldsymbol{A}}$ can be sampled from the continuous solution. In the continuous relaxation, the budget constraint becomes $\mathbb{E}[\text{Bernoulli}(\boldsymbol{B}')] = \sum \boldsymbol{B}'_{ij} \leq \Delta$, which can be dealt with by using projected gradient descent (Xu et al., 2019). Note that a continuous $\boldsymbol{B}'$ gives rise to a continuous $\tilde{\boldsymbol{A}}' \in [0,1]^{n \times n}$, whose elements $\tilde{\boldsymbol{A}}'_{ij}$ can be interpreted as the probability an edge $(i, j)$ being in $\tilde{\boldsymbol{A}}$. For large graphs, updating all entries in $\boldsymbol{B}'$ at once becomes infeasible. Projected Randomized Block Coordinate Descent (PRBCD) solves this by optimizing over sampled random blocks of limited size (Geisler et al., 2021).

### 2.2 Graph Transformers

Graph transformers (GTs) apply the popular transformer architecture for sequences (Vaswani et al., 2017) to arbitrary graphs. A general GT architecture is depicted in Fig. 2. In this work, we focus on GTs that apply global self-attention, where each node can attend to all other nodes. A "vanilla" structure-unaware self-attention head is defined as:

$$\text{Attn}(\boldsymbol{H}) = \text{softmax}\left(\frac{(\boldsymbol{H}\boldsymbol{W}_q)(\boldsymbol{H}\boldsymbol{W}_k)^\top}{\sqrt{d}}\right)(\boldsymbol{H}\boldsymbol{W}_v) \tag{3}$$

where $\boldsymbol{W}_q, \boldsymbol{W}_k, \boldsymbol{W}_v \in \mathbb{R}^{d \times d}$ are the weights for the *query*, *key*, and *value* projections. The individual attention scores can thus be defined as:

$$\alpha_{ij} = \text{softmax}(w_{ij}) = \frac{e^{w_{ij}}}{\sum_k e^{w_{ik}}}, \quad w_{ij} = \frac{\boldsymbol{W}_q^\top \boldsymbol{h}_i \cdot \boldsymbol{W}_k^\top \boldsymbol{h}_j}{\sqrt{d}} \tag{4}$$

Since this update is independent of the graph structure, many GTs apply a modified attention mechanism that also depends on the adjacency matrix. Additionally, a crucial and most common way to add structural information is by adding Positional Encodings (PEs) to the node features:

$$\boldsymbol{H}^{(0)} = \boldsymbol{X} + \psi(\boldsymbol{A}) \tag{5}$$

where $\psi$ represents the positional encoding of choice. Some architectures append $\psi(\boldsymbol{A})$ to $\boldsymbol{X}$ instead of summing, or otherwise jointly process $\boldsymbol{X}$ and $\psi(\boldsymbol{A})$. We categorize PEs roughly into three main categories: (1) distance encodings, (2) spectral encodings, and (3) random walk encodings. Some works distinguish between Structural Encodings (SEs) and PEs, where the term SE is used for encodings that make the GT aware of graph structure and the term PE for making a node aware of its relative position. However, there is no formal distinction between SEs and PEs (Müller et al., 2024) and for the purpose of this work, we do not semantically distinguish between SEs and PEs and use the term positional encoding to refer to any encoding based on $\boldsymbol{A}$. Next, we describe the PEs and attention mechanisms of the five representative GT models that we attack. For a detailed overview and taxonomy of current GTs we refer to Müller et al. (2024).

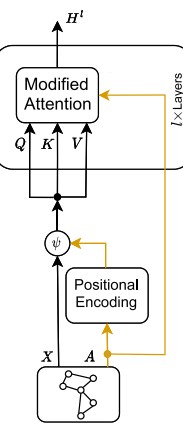

Figure 2: A generic graph transformer.

**Graphormer (Ying et al., 2021).** For the PEs, a degree embedding vector $\boldsymbol{z}_d \in \mathbb{R}^d$ is learned for each discrete node degree value $d$. The embeddings are added to the node features according to the node degrees:

$$\boldsymbol{h}_i^{(0)} = \boldsymbol{x}_i + \boldsymbol{z}_{\mathsf{deg}(v_i)} \tag{6}$$

Similarly, a learnable scalar $b_s \in \mathbb{R}$ is assigned to each discrete Shortest Path Distance (SPD) $s \in \mathbb{N}_0$. This value is added to the raw attention scores and results in a re-weighting of the attention weights between two nodes based on their distance in the graph:

$$\hat{w}_{ij} = w_{ij} + b_{\mathsf{spd}(v_i, v_j)}, \quad \alpha_{ij} = \mathsf{softmax}(\hat{w}_{ij}) \tag{7}$$

where $w_{ij}$ is set following Eq. 4. For graph-level tasks, a virtual node is added to the graph with its own distinct learnable bias $b_{\mathsf{virtual}}$, which is used as graph representation in the pooling stage.

**Spectral Attention Network (SAN)** (Kreuzer et al., 2021). SAN uses learned (spectral) Laplacian-based PEs that are based on the eigen-decomposition of the Laplacian $\boldsymbol{L}_{sym} = \boldsymbol{U}\boldsymbol{\Lambda}\boldsymbol{U}^\top$, where the diagonal entries of $\boldsymbol{\Lambda}_{ii} = \lambda_i$ are the eigenvalues of $\boldsymbol{L}_{sym}$ in ascending order $\lambda_1 \leq \lambda_2 \leq ... \leq \lambda_n$, and the columns of $\boldsymbol{U}$ are the corresponding eigenvectors. The PEs are computed by a learned transformer encoder that takes the $k$ smallest eigenvalues, which we denote by $\boldsymbol{\Lambda}_k \in \mathbb{R}^{k \times k}$, and their corresponding eigenvectors $\boldsymbol{U}_k \in \mathbb{R}^{n \times k}$ as input. Concretely, for each node $v_i$, its PEs are initialized as the concatenation of the eigenvalues and the $i$-th row of $\boldsymbol{U}_k$:

$$\boldsymbol{P}_i = [\mathsf{diag}(\boldsymbol{\Lambda}_k) \, \| \, (\boldsymbol{U}_k)_i] \in \mathbb{R}^{k \times 2} \tag{8}$$

Further processing by a transformer encoder results in $\boldsymbol{p}_i = f(\boldsymbol{P}_i) \in \mathbb{R}^{d_p}$, which is concatenated to the node features: $\boldsymbol{h}_i^{(0)} = \boldsymbol{x}_i \, \| \, \boldsymbol{p}_i$. Regarding the main graph transformer attention mechanism, it is modified to have two separate key and query weights for connected and unconnected node-pairs. The attention scores to the connected nodes and to the unconnected nodes are computed independently, each with a softmax. A hyperparameter $\gamma \in \mathbb{R}^+$ controls how the two scores are relatively scaled, varying the bias towards sparse or full attention:

$$\alpha_{ij} = \begin{cases} \frac{1}{1+\gamma}\mathsf{softmax}_{\mathcal{N}_i}\left(\boldsymbol{W}_{q,\mathsf{real}}^\top \boldsymbol{h}_i \cdot \boldsymbol{W}_{k,\mathsf{real}}^\top \boldsymbol{h}_j / \sqrt{d}\right) & \text{if } (v_i, v_j) \text{ is a real edge} \\ \frac{\gamma}{1+\gamma}\mathsf{softmax}_{\mathcal{V}\backslash\mathcal{N}_i}\left(\boldsymbol{W}_{q,\mathsf{fake}}^\top \boldsymbol{h}_i \cdot \boldsymbol{W}_{k,\mathsf{fake}}^\top \boldsymbol{h}_j / \sqrt{d}\right) & \text{otherwise} \end{cases} \tag{9}$$

where $\mathcal{N}_i$ is the first-order neighbors of node $v_i$ (including $v_i$).

**Graph Inductive Bias Transformer (GRIT)** (Ma et al., 2023). GRIT's PEs are based on random walk probability matrices for walks of lengths 0 to $k-1$. Concretely, the PEs are based on a 3D tensor:

$$\boldsymbol{P} = [\boldsymbol{I}, \boldsymbol{M}, \boldsymbol{M}^2, ..., \boldsymbol{M}^{k-1}] \in \mathbb{R}^{n \times n \times k}, \quad \text{with} \quad \boldsymbol{M} = \boldsymbol{D}^{-1}\boldsymbol{A} \tag{10}$$

This yields an embedding vector $\boldsymbol{P}_{ij} \in \mathbb{R}^k$ for each of the $n^2$ node-pairs $(v_i, v_j)$. The diagonal vector entries are transformed to dimension $d$ by a linear layer and added to the node features as PEs: $\boldsymbol{h}_i^{(0)} = \boldsymbol{x}_i + g_1(\boldsymbol{P}_{ii})$. Additionally, all $n^2$ vectors are transformed by a separate linear layer and added as node-pair features: $\boldsymbol{h}_{i,j}^{(0)} = g_2(\boldsymbol{P}_{ij})$. The node representations $\boldsymbol{h}_i$ and node-pair representations $\boldsymbol{h}_{i,j}$ are updated in each transformer layer by a modified attention mechanism, which includes an adaptive degree-scaler that is applied to the node representations:

$$\boldsymbol{h}_{i,update} = (\boldsymbol{h}_i \odot \boldsymbol{\theta}_1) + \log(1 + \deg(v_i)) \cdot (\boldsymbol{h}_i \odot \boldsymbol{\theta}_2) \tag{11}$$

where $\boldsymbol{\theta}_1, \boldsymbol{\theta}_2 \in \mathbb{R}^d$ are learnable weights.

**General, Powerful, Scalable (GPS) Graph Transformer** (Rampášek et al., 2022). GPS is a modular framework that first consists of a positional encoding of choice that is concatenated to the node features and again processed with an MLP before being passed through $L$ GPS layers. Each GPS layer combines the local message passing of an MPNN with a global attention update as follows:

$$f_{\mathsf{GPS}}(\boldsymbol{H}, \boldsymbol{A}) = \mathsf{MLP}\left(f_{\mathsf{MPNN}}(\boldsymbol{H}, \boldsymbol{A}) + f_{\mathsf{GlobalAttn}}(\boldsymbol{H})\right) \tag{12}$$

There are several different choices of PEs, MPNNs, and global attention mechanisms tested by Rampášek et al. (2022). We consider the configuration with (spectral) Laplacian PEs that are encoded using DeepSet (Zaheer et al., 2017) (compared to a transformer encoder in SAN), local GatedGCN (Bresson & Laurent, 2018) as an MPNN, and standard transformer global attention (see Eq. 3), as this configuration choice is the most common one by Rampášek et al. (2022).

**Polynormer** (Deng et al., 2024). In the Polynormer model, the input is first processed by $L_{local}$ local message passing layers and then by $L_{global}$ global attention layers. Notably, PEs are not used. For both type of layers, the node representation update is defined by a second-degree polynomial equation of the form (omitting normalizations and activation functions):

$$\boldsymbol{H}_{update} = (\boldsymbol{S}\boldsymbol{H}\boldsymbol{W}_v) \odot \left(\boldsymbol{H}\boldsymbol{W}_p + \sigma\left(\mathbf{1}\boldsymbol{\beta}^\mathsf{T}\right)\right), \quad \text{with attention matrix} \quad \boldsymbol{S} = \boldsymbol{S}(\boldsymbol{A}, \boldsymbol{H}) \tag{13}$$

where $\sigma$ is the sigmoid function and each layer has learnable weights $\boldsymbol{W}_v, \boldsymbol{W}_p \in \mathbb{R}^{d \times d}$ and $\boldsymbol{\beta} \in \mathbb{R}^d$. To calculate $\boldsymbol{S}$, for local layers, the local attention mechanism from GAT (Veličković et al., 2018) is used. For global layers, a linearized global attention mechanism is used based on the kernel trick. Instead of computing the softmax after the query-key multiplication (Eq. 4), an element-wise sigmoid function is applied separately to the queries and keys. Then a simple row-wise normalization ensures that rows sum to one:

$$\alpha_{ij} = \frac{w_{ij}}{\sum_k w_{ik}}, \quad w_{ij} = \sigma\left(\boldsymbol{W}_q^\mathsf{T}\boldsymbol{h}_i\right) \cdot \sigma\left(\boldsymbol{W}_k^\mathsf{T}\boldsymbol{h}_j\right) \tag{14}$$

where $\boldsymbol{W}_q, \boldsymbol{W}_k$ are the query and key projection matrices. Since the nonlinearity is applied before the multiplication, the order of operations can be changed to avoid computing the full attention matrix $\boldsymbol{S}$, reducing complexity from $O(n^2 d)$ to $O(nd^2)$. Thus, Polynormer is a type of GT that achieves *linear complexity* in the number of nodes.

## 3 Attacking Graph Transformers

The main obstacles for gradient-based structure attacks on GTs are PEs and attention mechanisms that are designed to operate on the discrete graph structure. As a result, even if one wants to solve the associated optimization problem in Eq. 1 for a relaxed continuous adjacency matrix $\tilde{\boldsymbol{A}}'$, the GT model $f_\theta$ is often discontinuous and non-differentiable w.r.t. $\tilde{\boldsymbol{A}}'$, making continuous optimization through gradient-based attacks inapplicable. Thus, to enable obtaining useful gradients, we need to relax the structure-aware components such as PEs and specialized attention mechanisms in $f_\theta$, giving rise to a relaxed GT model $\tilde{f}_\theta$. For designing effective continuous relaxations that lead to a useful $\tilde{f}_\theta$, we identify three main principles:

> **Principle I: Relaxed and target models should coincide for discrete inputs.** The prediction should equal $\tilde{f}_\theta(\boldsymbol{A}) = f_\theta(\boldsymbol{A})$ for any discrete adjacency matrix $\boldsymbol{A} \in \{0, 1\}^{n \times n}$.
>
> **Principle II: $\tilde{f}_\theta$ can interpolate between any different discrete graphs.** In other words, $\tilde{f}_\theta(\tilde{\boldsymbol{A}}')$ should be continuous w.r.t. $\tilde{\boldsymbol{A}}'$, and it should be differentiable almost everywhere w.r.t. $\tilde{\boldsymbol{A}}'$.
>
> **Principle III: The relaxed model $\tilde{f}_\theta$ must be efficient.** It is a critical property that the relaxation does not excessively increase the memory and runtime complexity.

We do not require continuous differentiability in *Principle II*, as to obtain informative gradients w.r.t. the input data to effectively optimize the attack loss, we do not need to enforce stronger standards on $\tilde{f}_\theta$ than imposed by the perhaps most widely used activation function ReLU, which is also continuous and differentiable almost everywhere except at 0. We discuss Principle II and the differentiability requirement in more detail in § G.1. Now, below, we develop continuous relaxations for several common GTs that follow the above outlined principles and thereby, enable the effective application of state-of-the-art gradient-based graph structure attacks as described in § 5. Next to covering commonly used GT components, the following derivations should act as guiding examples on how to instantiate the above principles to develop effective continuous relaxations that enable strong adaptive attacks for a GT architecture or component of choice.

**Graphormer**. The degree PEs $\boldsymbol{z}_{\mathsf{deg}(v_i)}$ in Eq. 6 and SPD biases $b_{\mathsf{spd}(v_i, v_j)}$ in Eq. 7 are indexed by the discrete values of the node degrees (# of neighbors) and shortest path distances (# of hops). To enable the use of continuous degrees, we define a linear interpolation between the PE vectors of the two closest integer degree values:

$$\tilde{\boldsymbol{z}}_{\mathsf{deg}(v_i)} = \eta \cdot \boldsymbol{z}_{\mathsf{d}_l+1} + (1 - \eta) \cdot \boldsymbol{z}_{\mathsf{d}_l}$$
$$\text{with} \quad \mathsf{d}_l = \lfloor \mathsf{deg}(v_i) \rfloor, \quad \text{and} \quad \eta = \mathsf{deg}(v_i) - \mathsf{d}_l \tag{15}$$

Increasing the edge probabilities to a node also increases the expected discrete degree. However, the edge probabilities are more challenging to interpret for the SPDs. When a very small edge probability lies on a (simple) shortest path, the path is less likely to exist in the discrete sampled adjacency matrix. Therefore, low edge probabilities should only marginally affect the original SPDs. To model this relationship, we use the reciprocal of the adjacency matrix $\boldsymbol{R}_{ij} = 1/\bar{\boldsymbol{A}}'_{ij}$ to find continuous proxy shortest path distances $\mathsf{rspd}_{ij} = \mathsf{spd}(v_i, v_j | \boldsymbol{R})$. We interpolate between the closest discrete values again and obtain:

$$\tilde{b}_{\mathsf{spd}(v_i, v_j)} = \eta \cdot b_{\mathsf{s}_l+1} + (1 - \eta) \cdot b_{\mathsf{s}_l}$$
$$\text{with} \quad \mathsf{s}_l = \lfloor \mathsf{rspd}_{ij} \rfloor, \quad \text{and} \quad \eta = \mathsf{rspd}_{ij} - \mathsf{s}_l \tag{16}$$

Note that for discrete edge probability values 0 and 1, the reciprocal edge weights become $-\infty$ and 1, respectively, yielding the original SPDs. Hence, we do not alter the clean predictions if $\delta\boldsymbol{A} = \boldsymbol{B} = \boldsymbol{0}$ (see *Principle I*).

**SAN**. We first discuss how to relax SAN's attention mechanism before discussing how to tackle relaxing the spectral PEs. To allow for a smooth transition between SAN's two separate sparse attention mechanisms in Eq. 9, we formally convert both to full global attention. Then, the goal of the relaxed sparse attention mechanisms is to properly weigh the contribution an edge has on the attention score calculation by the probability $\tilde{\boldsymbol{A}}'_{ij}$ that the corresponding edge is in the final perturbed adjacency matrix. This can be achieved by adding the log-probabilities of the edges belonging to one of the attention mechanisms in Eq. 9 to the attention logits, where the corresponding probabilities are $p_{ij} = \tilde{\boldsymbol{A}}'_{ij}$ for the sparse attention originally defined over $\mathcal{N}_i$, and $p_{ij} = 1 - \tilde{\boldsymbol{A}}'_{ij}$ for the sparse attention originally defined over $\mathcal{V} \setminus \mathcal{N}_i$, as one obtains

$$\tilde{w}_{ij} = w_{ij} + \log(p_{ij})$$
$$\tilde{\alpha}_{ij} = \mathsf{softmax}(\tilde{w}_{ij}) = \frac{e^{\tilde{w}_{ij}}}{\sum_k e^{\tilde{w}_{ik}}} = \frac{p_{ij} \cdot e^{w_{ij}}}{\sum_k p_{ik} \cdot e^{w_{ik}}} \tag{17}$$

As a result, for a discrete connected edge $\tilde{\boldsymbol{A}}'_{ij} = 1$, the log-probabilities become 0, or $-\infty$ respectively. Thus, such an edge still fully contributes to the connected attention mechanism (over $\mathcal{N}_i$), while not affecting the

disconnected one (over $\mathcal{V} \setminus \mathcal{N}_i$) – and vice versa for disconnected edges with $\tilde{\boldsymbol{A}}'_{ij} = 0$. Thus, the relaxation in essence still is a sparse attention mechanism and the discrete output remains unchanged (see *Principle I*), while interpolating smoothly between any different discrete graphs (see *Principle II*). Some care has to be taken so that the gradient computation w.r.t. $\tilde{A}'_{ij}$ is numerically stable, which we discuss in § G.2. Here we want to note that the so-developed sparse attention relaxation can be generally applied to any sparse attention mechanism used in other GTs, which we demonstrate when deriving a relaxed Polynormer model at the end of this section.

Now, regarding the spectral PEs, note that the Laplacian matrix itself is a continuous function of the entries in the adjacency matrix. However, its eigen-decomposition used for the PEs poses some challenges for gradient computation, especially w.r.t. the eigenvectors. The problems arise because: (a) the choice of direction (sign) for eigenvectors is arbitrary, (b) the choice of an eigenvector-basis of the eigenspace of a repeated eigenvalue is arbitrary, thus the gradient is not well defined, (c) for eigenvalues that are close together, the corresponding eigenvector gradients are numerically unstable. To avoid direct gradient computation, we use results from matrix perturbation theory (Stewart & Sun, 1990; Bamieh, 2022) to approximate the perturbed eigen-decomposition as a simpler function of the input perturbation. We define the perturbation on the Laplacian as $\delta \boldsymbol{L}_{sym} = \tilde{\boldsymbol{L}}_{sym} - \boldsymbol{L}_{sym}$, where $\tilde{\boldsymbol{L}}_{sym}$ is the Laplacian of the perturbed continuous adjacency matrix $\tilde{\boldsymbol{A}}'$. Following Bamieh (2022), the first-order approximations for the eigenvalues and eigenvectors are:

$$\tilde{\boldsymbol{\Lambda}} = \boldsymbol{\Lambda} + \delta \boldsymbol{\Lambda}, \quad \delta \boldsymbol{\Lambda} \approx \mathsf{diag}(\boldsymbol{U}^\mathsf{T} \delta \boldsymbol{L}_{sym} \, \boldsymbol{U}) \tag{18}$$

$$\tilde{\boldsymbol{U}} = \boldsymbol{U} + \delta \boldsymbol{U}, \quad \delta \boldsymbol{U} \approx -\boldsymbol{U} \, \left(\boldsymbol{\Pi} \odot \left(\boldsymbol{U}^\mathsf{T} \delta \boldsymbol{L}_{sym} \, \boldsymbol{U}\right)\right) \tag{19}$$

$$\text{with} \;\; \boldsymbol{\Pi}_{ij} = \begin{cases} \frac{1}{\lambda_i - \lambda_j} & \text{if } \lambda_i \neq \lambda_j \\ 0 & \text{else} \end{cases}$$

However, when repeated eigenvalues are present in the unperturbed Laplacian, special care for the choice of the eigenvectors in $\boldsymbol{U}$ that span the eigenspaces of the repeated eigenvalues is required. This case is treated by Bamieh (2022) and we show the application to our case in § F.1. A different strategy used by e.g. Lin et al. (2022), consists of adding a bit of random noise to $\tilde{\boldsymbol{L}}_{sym}$ in hopes of breaking apart any repeated eigenvalues, such that is possible to directly backpropagate through the eigen-decomposition. We elaborate on this strategy and propose our own alternative in § F.2.

**GRIT.** Relaxing the perturbed adjacency matrix $\tilde{\boldsymbol{A}}$ to be used in Eq. 10 to be continuous and allowing for fractional node degrees in Eq. 11, one can see that the so relaxed GRIT model fulfills all three main principles for continuous relaxations outlined above. Thus, as only GT architecture, GRIT and its used random walk embeddings do not require special treatment beyond relaxing $\tilde{\boldsymbol{A}}$ to be continuous, to yield an effective continuous relaxation.

**GPS.** As GPS' Laplacian PEs only differ to SAN's by using a DeepSet encoder instead of a transformer encoder, we can use the same relaxation as derived for SAN in Eqs. 18 & 19. Note that Rampášek et al. (2022) lists six different categories of PEs, which they group into local, global, and relative positional encodings and local, global, and relative structural encodings. Most of the example encodings provided for each category are based on distances (shortest paths), node degrees, spectral decomposition, or random walks. Thus, they can be directly tackled by the relaxations developed for Graphormer (Eqs. 6 & 7), SAN (Eqs. 18 & 19), or GRIT. This highlights the common and widespread usage of distance, spectral, or random-walk encodings and thus, the broad applicability of our developed PE relaxations to other GT models.

Now, we turn to the GPS layers defined by Eq. 12. The global attention $f_{\text{GlobalAttn}}$ requires no modification, as it does not depend on the adjacency. However, we need to take a closer look at $f_{\text{MPNN}}$ for which GPS uses a GatedGCN. The update of the local GatedGCN step for the embedding of a node $i$ is defined as:

$$\boldsymbol{h}_{i,update} = \boldsymbol{h}_i + \rho \left( \boldsymbol{W}_{n1} \boldsymbol{h}_i + \frac{1}{\boldsymbol{n}_i} \odot \sum_{v_j \in \mathcal{N}_i \setminus \{v_i\}} \boldsymbol{\eta}_{ij} \odot \boldsymbol{W}_{n2} \boldsymbol{h}_j \right), \quad \boldsymbol{n}_i = \sum_{v_j \in \mathcal{N}_i \setminus \{v_i\}} \boldsymbol{\eta}_{ij} \tag{20}$$

$$\text{with edge gates} \quad \boldsymbol{\eta}_{ij} = \sigma \left( \boldsymbol{W}_{e1} \boldsymbol{h}_i + \boldsymbol{W}_{e2} \boldsymbol{h}_j \right)$$

where $\boldsymbol{W}_{n1}, \boldsymbol{W}_{n2}, \boldsymbol{W}_{e1}, \boldsymbol{W}_{e2}$ are the learnable parameters and $\rho$ is the activation function. The edge gates $\boldsymbol{\eta}_{ij} \in \mathbb{R}^d$ can be interpreted as per-dimension local attention weights and control the aggregation strength

from each neighbor. Thus, to achieve a continuous relaxation, we can define that a node $v_j \in \mathcal{N}_i$ if $\tilde{\boldsymbol{A}}'_{ij} > 0$ and scale the edge gates by the probability that the neighbor node is connected:

$$\tilde{\boldsymbol{\eta}}_{ij} = p_{ij} \; \boldsymbol{\eta}_{ij}, \quad p_{ij} = \tilde{\boldsymbol{A}}'_{ij} \tag{21}$$

This is in line with the way how other MPNNs such as a traditional GCN are relaxed to allow for gradient-based structure attacks (Geisler et al., 2021).

**Polynormer.** The global attention mechanisms defined in Eq. 14 does not depend on the adjacency. Thus, relevant to deriving a relaxed model is how the attention matrix $\boldsymbol{S}$ in Eq. 13 is calculated for the local layers based on the local (sparse) attention mechanism from GAT. For this, recognize that the $i$-th row of the result of the matrix product $\boldsymbol{SHW}_v$ that concerns the intermediate embedding $\tilde{\boldsymbol{h}}_{i,update}$ for a node $i$ after sparse aggregation and before the application of the element-wise multiplication in Eq. 13, can be written as:

$$\boldsymbol{h}_{i,update} = \rho \left( \sum_{v_j \in \mathcal{N}_i} \alpha_{ij} \boldsymbol{W}_v^\mathsf{T} \boldsymbol{h}_j \right), \quad \alpha_{ij} = \mathsf{softmax}_{\mathcal{N}_i}(w_{ij}), \quad w_{ij} = \rho \left( \boldsymbol{a}_s^\mathsf{T} \; \boldsymbol{W} \boldsymbol{h}_i + \boldsymbol{a}_t^\mathsf{T} \; \boldsymbol{W} \boldsymbol{h}_j \right) \tag{22}$$

where $\rho$ is an activation function and $\boldsymbol{W}, \boldsymbol{a}_s, \boldsymbol{a}_t$ are the learnable parameters. Now, structure information enters the sparse attention computation for $\alpha_{ij}$ through attending only to the neighbors $\mathcal{N}_i$. To relax this sparse attention mechanism, we can use the same continuous relaxation derived for SAN's sparse attention in Eq. 9 by formally converting the sparse $\alpha_{ij}$ computation (and summation) in Eq. 22 to full attention and adding $\log p_{ij}$ as bias to the attention logits $w_{ij}$ with $p_{ij} = \boldsymbol{A}'_{ij}$. This highlights the general applicability of the relaxed sparse attention developed in Eq. 9 for SAN to other sparse attention mechanisms deployed in other GT models. Note that in contrast to relaxing an MPNNs aggregation through scaling each summation (aggregation) term by $\tilde{\boldsymbol{A}}'_{ij}$, the scaling for the relaxed sparse attention is already included in the $\alpha_{ij}$ computation. We close this section by noting that Polynormer does not use PEs and hence, we do not need to develop a relaxation for them.

**Computational Complexity.** All our relaxations adhere to Principle III of efficiency and do not excessively increase runtime or memory complexity. In particular, the relaxations do not affect the asymptotic time complexities w.r.t. the number of nodes or edges of the models. We discuss the computational complexities of our relaxations in more detail in § G.3.

## 4 Node Injection Attack

We also consider the relevant case of inserting nodes into an existing graph structure. In contrast to the usual framing of Node Injection Attack (NIA), where the attacker also chooses the node features for the new *vicious* nodes (Wang et al., 2020), we connect existing nodes from other graphs of an inductive graph dataset (excluding their own connections in the other graph). Therefore, the nodes' features are fixed but physically realizable even if, e.g., they represent embeddings of natural language. This alleviates us from a somewhat subjective definition of imperceptibility required to craft the node features in the existing NIA. Hence, our attack solely focuses on "structure" perturbations and their influence on the PEs, which are of particular interest for attacking GTs.

We formulate our node injection attack as a structure attack on an augmented graph that includes both the original nodes and the set of potential injection nodes. This formulation enables the use of the same PRBCD attack optimization, where the edge flip budget constraint also serves as an upper bound for the number of nodes that can be injected. We provide the details of extending PRBCD to node injection in § A.

**Node probability for smooth node insertion**. The continuous optimization of structure attacks in § 2.1 assigns probabilities to edges-flips, while nodes are assumed to be part of the graph. In contrast, during NIAs nodes also have certain probabilities of being included. To approximate these node probabilities from the edge weights in a general way, we propose a simple iterative computation. We can calculate the probability $p_i$ of $v_i$ being connected to the graph, by using the probability of being connected to its neighbors and the probabilities that these neighbors themselves are connected to the graph. We start with the assumption that

all nodes are connected to the graph and update using the edge probabilities:

$$p_i^{(t+1)} = 1 - \prod_{v_j \in \mathcal{N}_i \setminus \{v_i\}} (1 - \tilde{\boldsymbol{A}}_{ij}' \cdot p_j^{(t)}), \quad \text{with} \quad p_i^{(0)} = 1 \tag{23}$$

An illustrative example is shown in § A.1. To ensure that the model output is continuous w.r.t. node injections, this node probability is used to compute a weighted sum or mean in the graph-pooling for graph level tasks. Additionally for GTs, we use the node probability to bias the global pairwise attention scores which result in a continuous weighting of the attention scores analogous to Eq. 17, where the bias probability is set to the node probability $p_{ij} = p_j$. Notably, this node probability bias can be applied to any global attention mechanism even when it originally does not depend on the adjacency matrix (as is the case for e.g. GPS and Polynormer).

## 5 Evaluation

In what follows we describe the experimental details for our reported results. The code to reproduce our results can be found at `https://github.com/isefos/gt_robustness`.

**Datasets**. We first evaluate our structure attacks on *CLUSTER* (Dwivedi et al., 2023) which contains SBM-generated graphs with 6 clusters. A single node in each cluster is labeled and the task is to predict the cluster of all nodes (inductive node classification). We also consider the graph classification dataset *Reddit Threads* (Rozemberczki et al., 2020). It contains many small graphs (without node features) that represent users that are connected if they directly reply to each other in the thread. The task is to predict whether the thread is discussion-based or not. For our node injection attacks, we evaluate on the *UPFD* fake news detection datasets (Dou et al., 2021). There are 2 datasets: *politifact*, with political; and *gossipcop* with celebrity fake news. The graphs consist of "retweet" trees, where each node contains the user features and the edges represent retweets. Additionally, the root nodes contain features related to the news content and are consequently not considered for node injection. We do not perturb the original graph structure and if node injection does not result in a tree structure, we take the maximum spanning tree to ensure that all perturbations are valid retweet trees. The task is binary classification of whether the graph contains fake news or not. Further details of the datasets and splits used are given in § B.

Due to GTs (usually) quadratic scaling in the number of nodes, their application is limited to smaller graphs. While GTs are most widely applied to molecule data, adversarial attacks are of little practical relevance in that domain. Thus, we omit molecular datasets from our evaluations.

**Models.** We investigate the five representative GT models for which we developed continuous relaxations in § 3: Graphormer, SAN, GRIT, GPS, and Polynormer. For their model training we do a hyperparameter search, choosing the model with the highest validation metric and we describe the hyperparamter search and the final hyperparameters used for the models in § E.

**Attacks**. As explained in § 2.1, we study *untargeted global evasion* attacks. *Global* applies to the task of node classification and means that our attacks try to decrease the overall accuracy of all (test) node predictions in the graph. This is more challenging than *local* attacks, which only attack a single victim node prediction such as Nettack (Zügner et al., 2018), which cannot be effectively used for a global attack. *Evasion* means the graph structure of the test input is modified for a trained model with fixed weights. This is different from *poisoning* attacks such as Mettack (Zügner & Günnemann, 2019), where the victim model is trained on the perturbed graph. *Poisoning* is much more relevant for transductive learning tasks (often on a single graph), for which GTs are rarely used.

We show results for 5 different attack types. *Adaptive PRBCD* uses our relaxations described in § 3 for a gradient-based PRBCD attack. *Random perturbation* is a simple baseline, where a single random perturbation of the adjacency matrix is used. In contrast, *random attack* is a brute-force random search that tests many random perturbations and selects the best. To match the computational budget of the adaptive attacks, it gets the same number of model evaluations. The *GCN PRBCD transfer* attack transfers the perturbation computed from attacking a GCN with an (adaptive) PRBCD attack, to the GT models. This is a relatively strong baseline attack that follows the same principle as many other established GNN attacks: it is a

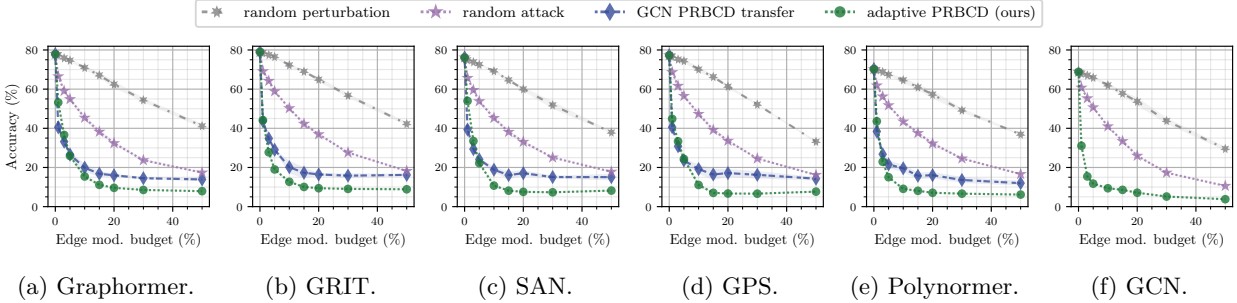

(a) Graphormer.  (b) GRIT.  (c) SAN.  (d) GPS.  (e) Polynormer.  (f) GCN.

Figure 3: Global structure (evasion) attack results for CLUSTER (inductive node classification).

gradient-based attack (PRBCD) on a simpler surrogate (GCN) that gets transferred to the victim model. Moreover, it is the main (global evasion) attack for non-GCN models proposed and used by Geisler et al. (2021), where it has been shown to be very effective against other GNNs. Finally, we study how the adaptive attacks for the individual five considered GTs transfer to one another. Fig. 1 shows the best out of all applied attacks (including the individual GT transfers) for each investigated model and dataset.

For all datasets, we evaluate our attacks on the 50 first graphs in the test set and report average and standard deviation over 4 random seeds. For UPFD node injection, we use a small block size of 1000, which is necessary due to the quadratic scaling of GTs. We optimize all our adaptive attacks for 125 steps and sample 20 discrete perturbations from the result, of which we take the strongest. For all other attack hyperparameters, we use default values that performed well in preliminary evaluations. For all main results, we use all of our continuous relaxations proposed in § 3. We report and discuss ablation results using different combinations of relaxation in § C.5.

# 6    Attack Results

In this section, we present the first principled analysis on the robustness of GTs on five representative architecture types (Graphormer, GRIT, SAN, GPS, Polynormer) enabled by our developed adaptive attacks based on our general principles for continuous relaxations outline in § 3. We define different goals for our evaluation: **(A)** efficacy of the proposed adaptive attacks, **(B)** providing an accurate assessment of GT robustness for relevant real-world tasks. To this end, we perform our evaluation on datasets with varying complexity. Towards **(A)** we explore the robustness of GTs on CLUSTER and Reddit Threads, which comprise simple, interpretable structures. This exploration helps us evaluate the effectiveness of the proposed relaxations, ideally leading our attacks to target semantically meaningful structures within the dataset. We address **(B)** through evaluations on UPFD. Here, we constrain our attack to remain within the predefined tree structure of the dataset. As a result, the attack represents impersonating an existing user who is retweeting the respective news article. This evaluation goes beyond previous robustness analyses of citation networks in GNNs (Zügner et al., 2018; Geisler et al., 2021), offering a more practical use case and semantically meaningful attacks (Hu et al., 2024; Wang et al., 2023b). We quantify and measure the semantic unnoticability of our attacks on the UPFD datasets in § C.6.

**CLUSTER**. Across all models, our adaptive attacks result in the strongest perturbations except for the smallest budgets, as shown in Fig. 3. The effectiveness of the *random perturbation* baseline indicates an inherent fragility of the data. Intuitively, since only a single node in each cluster is labeled, attacking these labeled nodes requires little budget and leads to strong attacks. We manually inspected the adaptive attack perturbations and confirmed that most edge modifications are connected to the labeled nodes. The strength of the transfer attacks also indicates that the straightforward nature of the task leads to the same type of semantically meaningful model-independent perturbations. This outcome positively indicates the effectiveness of our adaptive attacks **(A)**, as they consistently identify meaningful perturbations across all GTs. To avoid the natural fragility in the data, we also evaluate a constrained attack that prohibits modifying edges to the labeled nodes, for which results are shown in § C.1. In § C.7, we evaluate and show the efficacy of our adaptive attacks in a local attack setting. The worst-case perturbation results for all models, including GAT

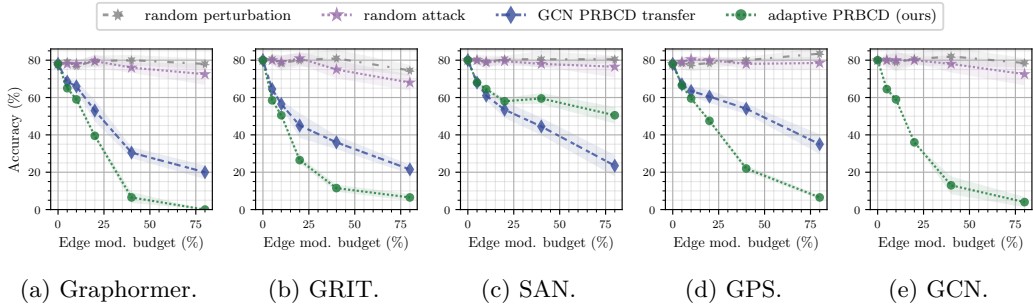

Figure 4: Structure (evasion) attack results for Reddit Threads (graph classification).

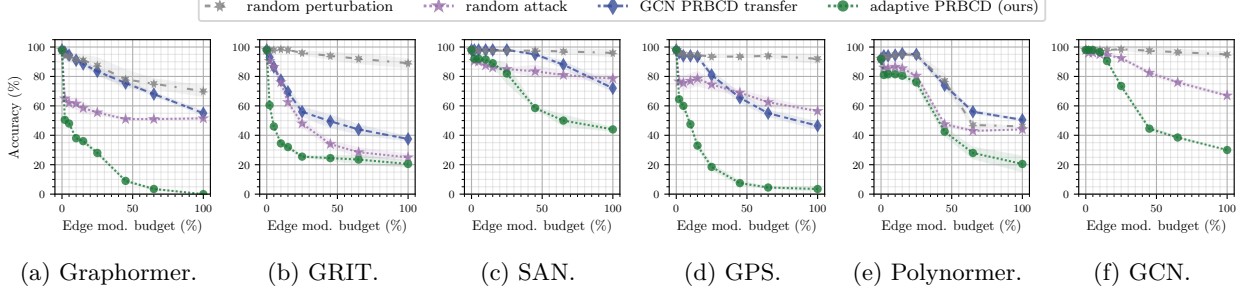

Figure 5: Node injection (evasion) attack results for UPFD gossipcop (graph classification).

and GATv2 (Brody et al., 2022), are shown in Fig. 1a. For these smaller budgets, the GTs are consistently more robust than the MPNNs, though their clean accuracies are also higher.

**Reddit Threads**. Fig. 4 shows that our adaptive attacks are significantly stronger than the baselines. SAN is the exception, where the adaptive attack is worse than transferring from GCN. This could be due to the perturbation approximation adding noise to the gradient updates, or because it results in a harder optimization function. For most models, the adversarial accuracy drops close to zero when up to 80% of the edges can be modified. This is likely because there are no node features and the prediction relies only on the graph structure. Interestingly, the random attacks never seem to work well, indicating that the gradient information provided by our relaxations is extremely helpful for finding good perturbations. Fig. 1b shows a comparison of the models' robustness for small budgets. While there are differences in robustness, all models follow a similar trend. Note that for this dataset we were unable to train a comparable Polynormer model. We hypothesize that this is related to Reddit Threads having no node features. While the other GTs explicitly include graph structure information through positional encodings, these are missing in Polynormer, which only implicitly considers graph structure through the sparse attention matrix obtained by employing a GAT. This may be insufficent graph inductive bias, if node features have no discriminative information.

**UPFD**. As shown in Fig. 5 for the gossipcop dataset, our adaptive attacks are significantly stronger than the baselines in most cases, providing the best estimates of the models' robustness. This highlights the efficacy and importance of our gradient-based adaptive attacks also for the node injection setting. In contrast to the results observed for the previous datasets, there are much more differences between models. Fig. 1 provides a direct model comparison of the worst-case perturbations for smaller budgets. It shows that the GCN model can exhibit considerably higher robustness than some GTs. The SAN model is the exception, as it is surprisingly robust for both UPFD datasets. These results reveal that GTs can showcase catastrophic vulnerabilities to adversarial modifications of the graph structure, even when these changes are constrained to meaningful perturbations. Results for the politifact dataset are shown in § C.2.

**Transferability**. For each of the five GT models, we collect the adversarial examples generated from our adaptive attack, and transfer them to the other GT models. In Fig. 6, we compare the strongest such transfer attack (*best transfer*) with the *GCN transfer* and *adaptive* attacks on UPFD gossipcop. The results show that our GT attack perturbations transfer better than from GCN. This may be because the GT models are more similar to each other than to a GCN. In some cases, *best transfer* is the overall strongest attack.

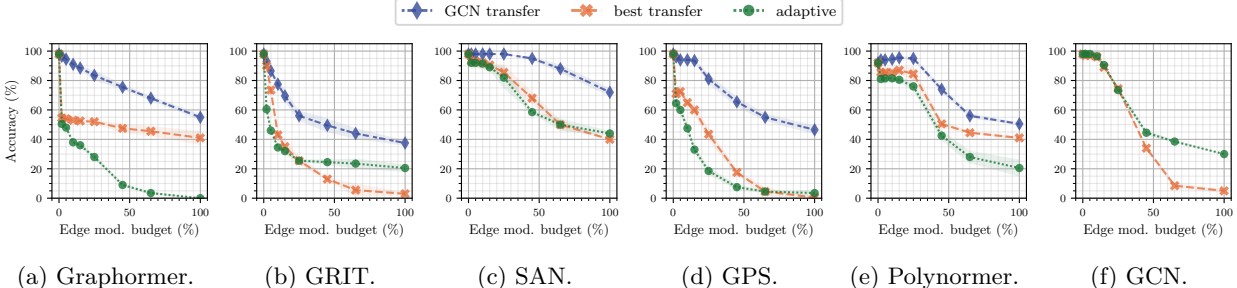

Figure 6: Transfer attack results (node injection, evasion) for UPFD gossipcop (graph classification).

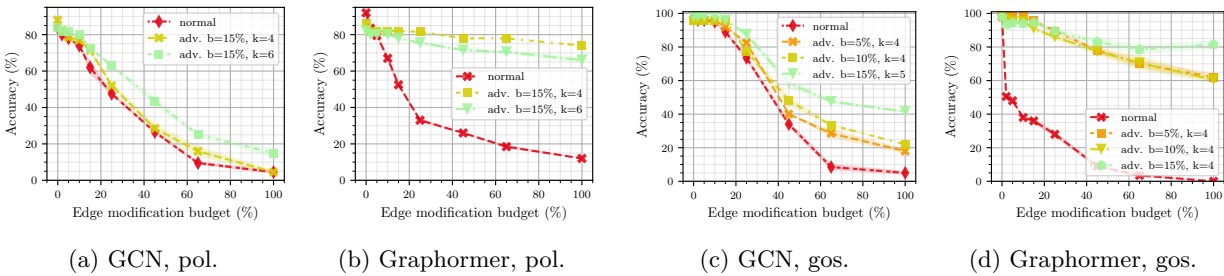

Figure 7: Node injection attack results for normally (red lines) and adversarially trained (orange, green, & yellow lines) GCN and Graphormer on UPFD politifact (pol) and gossipcop (gos) (graph classification).

However, note that choosing the best from up to eight (adaptively generated) attacks can be considered an ensemble with high computational cost. However, *best transfer* can be used as a "unit test" before laboriously designing adaptive attacks for a new GT architecture. Results of *best transfer* for other datasets and all *individual transfer results* across different GTs are available in § C.3 & § C.4 respectively.

## 7 Adversarial Training

We base our Adversarial Training (AT) implementation on the "Free" adversarial training of Shafahi et al. (2019). The main idea is to couple the attack and training optimizations by replaying the same mini-batch $k$ times. In each replay, both the attack perturbation and the model weights are updated. This approach allows us to apply stronger perturbations through multi-step optimization, while avoiding the overhead of only performing a single training update for every $k$ attack steps. In § D we provide the pseudocode and details our modifications to free AT, to make it applicable to our setting.

We explore adversarial training for Graphormer, one of the least robust models in our attack evaluation, and compare it to adversarially training a GCN as a baseline in a node injection attack scenario on the UPFD politifact dataset in Figs. 7a & 7b. Fig. 7a shows that a GCN struggles to benefit from adversarial training, while Fig. 7b demonstrates that AT significantly improves Graphormer's robustness, surpassing the adversarially trained GCN by a large margin. We find that these results are consistent across datasets with similar results on gossipcop in Figs. 7c & 7d. These findings show that the increased flexibility and capacity of graph transformers can offer significant advantages in learning robust models via adversarial training, even when the standard (non-adversarially trained) versions of these models are highly vulnerable, as we establish in § 6. These results complement and support the findings of Gosch et al. (2023a), who attribute the limited success of AT for traditional message-passing GNNs to the lack of their flexibility in adjusting their message-passing to adversarial examples. While Gosch et al. (2023a) establish that the capability of AT as a defense to structure perturbations can be significantly improved by making message-passing GNNs more flexible by making the graph filter in a graph convolution learnable, we show that breaking the static message passing by being able to learn to attend to nodes has a similar effect and is a key enabler for effective adversarial training.

# 8 Related Work

Triggered by the seminal works of Zügner et al. (2018); Dai et al. (2018), a research area emerged spanning attacks, defenses, and certification of message-passing GNNs (Jin et al., 2021; Günnemann, 2022; Guerranti et al., 2023; Gosch et al., 2023b; Sabanayagam et al., 2025). However, GTs have been entirely neglected despite being a very active field of research with demonstrated success on common benchmarks (Müller et al., 2024). Zhu et al. (2024) is the sole exception acknowledging this gap. However, they propose their own transformer-inspired defense component and evaluate it using transfer poisoning attacks. Thus, they do not shine light on the robustness of the diverse set of GTs nor do they study adaptive attacks. Mujkanovic et al. (2022) shows that adaptive attacks are crucial to correctly evaluate the robustness of GNNs. This follows similar results from the vision domain (Tramèr et al., 2020; Carlini & Wagner, 2017; Athalye et al., 2018).

Next to adaptive-attack works, our attack is rooted in the GNN robustness literature. Xu et al. (2019) proposes the first Projected Gradient Descent (PGD) attack for discrete $L_0$ perturbations of the graph structure, with a focus on message-passing architectures. Geisler et al. (2021) extend this PGD with a randomization scheme to obtain the efficient (gradient-based) Projected Randomized Block Coordinate Descent (PRBCD) attack. Gosch et al. (2023a) extend PRBCD with local constraints to allow for semantically more meaningful attacks, which is conceptually related to our semantically meaningful node injection attack. Further important related works are Lin et al. (2022); Zhu et al. (2018); Bojchevski & Günnemann (2019), where the authors study similar approximations for perturbations on the eigen-decomposition of the graph Laplacian. Moreover, Wang et al. (2023a) attack message-passing architectures on the UPFD fake news detection using reinforcement learning. As an entry to Node Insertion Attacks (NIA), we refer to Wang et al. (2020); Zou et al. (2021).

# 9 Conclusion

We provide the first principled study into the adversarial robustness of graph transformers. Concretely, we provide effective and general guiding principles for designing adaptive attacks for GTs. Consequently, we study five representative graph transformers which use three of the most commonly used positional encodings: random-walk-based, distance-based, and spectral PEs; as well as common sparse-attention mechanisms. Thus, our developed continuous relaxations for these GT components can find broad application to other GT models. Furthermore, our study demonstrates that GTs can be catastrophically fragile in many settings and more robust in others. This diverse picture underlines the importance and need for adaptive attacks to reveal such nuanced robustness properties. While the comparison of GT's and traditional GNN's robustness w.r.t. the studied attacks does not allow for a conclusion about which architecture is superior in terms of robustness when applying normal training, our adaptive attacks allow to uncover a strong difference in their robust learning capabilities. Concretely, we show how to leverage our adaptive attacks for adversarial training with GTs and that doing so, due to the flexibility of GTs, they have the potential to significantly outperform static message-passing GNNs in their robust learning performance, alleviating one of the key limitations of classic GNNs. One interesting direction for future work could be to study the optimality of relaxations, and how to define, measure, and prove this property.

**Broader Impact Statement**

While the threat model of attacking fake news detection could have a negative societal impact, our methods are applicable mostly in a white-box setting and, therefore, are much more useful to those who are developing fake news detection to probe and improve the robustness of their models. If a model developer has access to the right tools, we are convinced that the information advantage outweighs the potential negative effects.

**Acknowledgements**

This paper has been supported by the DAAD programme Konrad Zuse Schools of Excellence in Artificial Intelligence, sponsored by the German Federal Ministry of Education and Research and by the German Research Foundation, grant GU 1409/4-1. Leo Schwinn gratefully acknowledges funding by the Deutsche Forschungsgemeinschaft (DFG, German Research Foundation) - project number 544579844.

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

## A    Node Injection Attack Details

Let $\mathcal{D} = \{\mathcal{G}_1, ..., \mathcal{G}_N\}$ be the dataset of all graphs, where each graph $\mathcal{G}_i = (\mathcal{V}_i, \mathcal{E}_i)$ has $n_i$ nodes $\mathcal{V}_i = \{v_{i,1}, ..., v_{i,n_i}\}$ and a corresponding node feature matrix $\boldsymbol{X}_i \in \mathbb{R}^{n_i \times d}$. The total number of nodes in the dataset is $n_{\mathcal{D}} = \sum n_i$. Let $\mathcal{G}_{atk}$ be the graph that is being attacked. We define the candidate set of injection nodes as the union of the nodes of all other graphs: $\mathcal{V}_{cs} = \bigcup_{\mathcal{G}_i \in \mathcal{D} \setminus \mathcal{G}_{atk}} \mathcal{V}_i$, which contains $n_{cs} = n_{\mathcal{D}} - n_{atk}$ nodes with the corresponding features $\boldsymbol{X}_{cs}$. It is of course possible to restrict this candidate set if it is not sensible or feasible to include all nodes.

We can augment the original (connected) graph $\mathcal{G}_{atk} = (\boldsymbol{A}_{atk}, \boldsymbol{X}_{atk})$ by adding the injection candidate set as isolated nodes:

$$\hat{\mathcal{G}}_{atk} = (\hat{\boldsymbol{A}}_{atk}, \hat{\boldsymbol{X}}_{atk}), \quad \hat{\boldsymbol{A}}_{atk} = \begin{bmatrix} \boldsymbol{A}_{atk} & \boldsymbol{0} \\ \boldsymbol{0} & \boldsymbol{0} \end{bmatrix} \in \{0,1\}^{n_{\mathcal{D}} \times n_{\mathcal{D}}}, \quad \hat{\boldsymbol{X}}_{atk} = \begin{bmatrix} \boldsymbol{X}_{atk} \\ \boldsymbol{X}_{cs} \end{bmatrix} \in \mathbb{R}^{n_{\mathcal{D}} \times d} \tag{24}$$

Edge-flip perturbations to this augmented adjacency matrix, $\breve{\boldsymbol{A}} = \hat{\boldsymbol{A}} + \delta\hat{\boldsymbol{A}}$, model both structure perturbations and node injections together. As in Eq. 2, the perturbation $\delta\hat{\boldsymbol{A}}$ can be expressed in terms of a binary edge flip matrix: $\breve{\boldsymbol{A}} = \hat{\boldsymbol{A}} + (\mathbf{1}_n \mathbf{1}_n^{\mathsf{T}} - 2\hat{\boldsymbol{A}}) \odot \hat{\boldsymbol{B}}$, where:

$$\hat{\boldsymbol{B}} = \begin{bmatrix} \boldsymbol{B} & \boldsymbol{E} \\ \boldsymbol{E}^{\mathsf{T}} & \boldsymbol{F} \end{bmatrix} \in \{0,1\}^{n_{\mathcal{D}} \times n_{\mathcal{D}}} \tag{25}$$

Note that the edge flip budget $\Delta$ is also an upper bound for the number of nodes that can be injected: $0 \leq n_{in} \leq \Delta$. Since the attack budget is usually much smaller than the size of the candidate set, i.e. $\Delta \ll n_{cs}$, the perturbed augmented graph $\breve{\mathcal{G}} = (\breve{\boldsymbol{A}}, \hat{\boldsymbol{X}})$ still mostly contains isolated nodes. Therefore, we prune away all disconnected components, which for the unperturbed graph simply reverts the augmentation: $\mathsf{prune}(\hat{\mathcal{G}}) = \mathcal{G}$. However, for a perturbed augmented graph, this results in the perturbed graph that we are seeking:

$$\tilde{\mathcal{G}} = \mathsf{prune}(\breve{\boldsymbol{A}}, \hat{\boldsymbol{X}}) = (\tilde{\boldsymbol{A}}, \tilde{\boldsymbol{X}}), \quad \tilde{\boldsymbol{A}} \in \{0,1\}^{\tilde{n} \times \tilde{n}}, \quad \tilde{\boldsymbol{X}} \in \mathbb{R}^{\tilde{n} \times d} \tag{26}$$

Here, $n_{in}$ is the number of injected nodes, and $\tilde{n} = n + n_{in}$ is the total number of nodes of the perturbed graph. The NIA objective can thus be written as:

$$\max_{\hat{\boldsymbol{B}} \text{ s.t. } ||\hat{\boldsymbol{B}}||_0 < \Delta} \mathcal{L}_{atk}(f_\theta(\tilde{\mathcal{G}})), \quad \text{with} \quad \tilde{\mathcal{G}} = \mathsf{prune}(\hat{\boldsymbol{A}} + (\mathbf{1}_n \mathbf{1}_n^{\mathsf{T}} - 2\hat{\boldsymbol{A}}) \odot \hat{\boldsymbol{B}}, \ \hat{\boldsymbol{X}}) \tag{27}$$

where, $f_\theta$ is the trained GNN and $\mathcal{L}_{atk}$ is a suitable attack loss.

**Edge block sampling**. To optimize the objective, we can apply the relaxation $\hat{\boldsymbol{B}}'_{ij} \in [0,1]$, as shown in § 2.1. In this case, PRBCD (Geisler et al., 2021) not only enables more efficient optimization, but setting a smaller block size is crucial to limit the number of connected injection nodes during optimization, since GTs complexity scales with $O(\tilde{n}^2)$. Moreover, random block edge sampling allows us to control which parts of $\hat{\boldsymbol{B}}$ in Eq. 25 can be changed, e.g. not sampling in $\boldsymbol{B}$ results in pure node injections without modifying edges in the original graph. For NIAs with large candidate sets, we only sample from $\boldsymbol{E}$, as sampling from the $n_{cs}^2$ entries of $\boldsymbol{F}$ results in using most of the budget on disconnected injection node pairs that are later pruned away.

### A.1    Node Probability Example

We provide an illustrative example in Fig. 8 of how the iterative node probability is applied. Each iteration of Eq. 23 can be thought of as a message passing step to update the node probability approximation based on the neighbors current approximations:

$$p_i^{(t+1)} = 1 - \prod_{v_j \in \mathcal{N}_i \setminus \{v_i\}} (1 - \tilde{\boldsymbol{A}}'_{ij} \cdot p_j^{(t)}), \quad \text{with} \quad p_i^{(0)} = 1$$

The number of iterations should be set in the order of expected longest chain of added injection nodes. Therefore, very few iterations (2-5) should suffice for most NIAs.

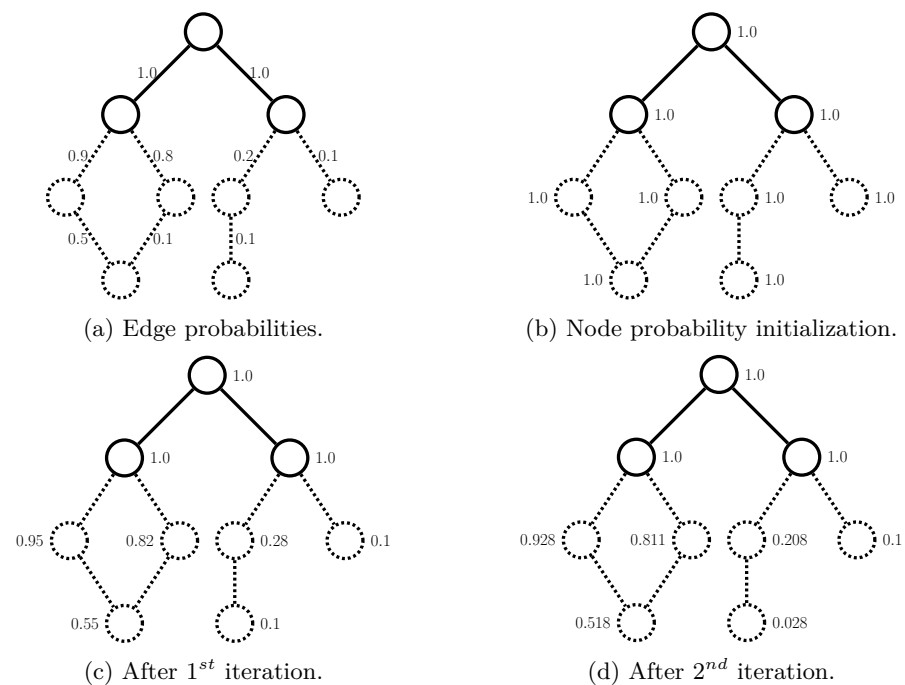

Figure 8: Node probability example. Dashed lines indicate injection nodes.

# B    Dataset Details

The inductive node classification dataset CLUSTER (Dwivedi et al., 2023) has 12 000 graphs with an average of 117.2 nodes. We used the standard PyG train/val/test split of 83.3/8.3/8.3% graphs. The binary graph classification dataset Reddit Threads (Rozemberczki et al., 2020) contains 203 088 graphs with an average of 23.9 nodes. We used a stratified random split of 75/12.5/12.5%. The binary graph classification dataset UPFD gossipcop (Dou et al., 2021) contains 5464 graphs with an average of 58 nodes. We use the standard PyG split of 20/10/70%. The binary graph classification dataset UPFD politifact (Dou et al., 2021) contains 314 graphs with an average of 131 nodes. We use the standard PyG split of 20/10/70%.

# C    Additional Attack Results

## C.1    CLUSTER Constrained Attack

Fig. 9 shows the attack results for the CLUSTER dataset when constraining edge perturbations such that edges to the labeled nodes cannot be flipped. As expected, this significantly reduces the attack strength compared to the unconstrained setting shown in Fig. 3.

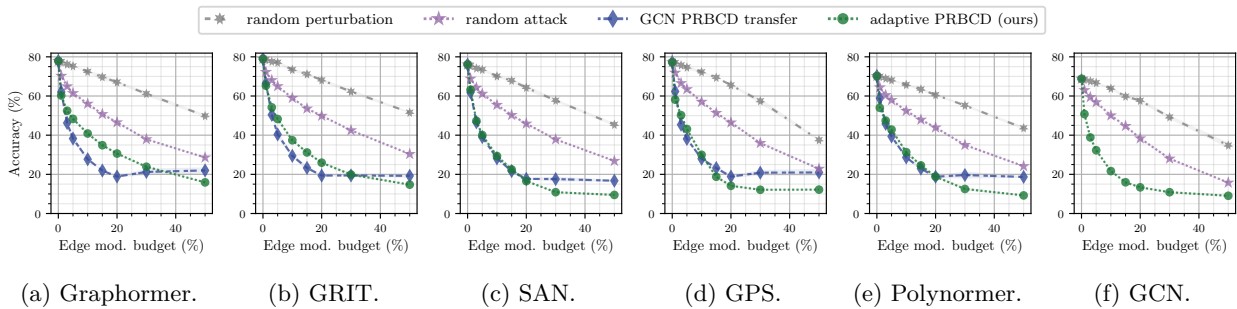

Figure 9: CLUSTER constrained attack results.

## C.2 UPFD Politifact Node Injection Attack

Fig. 10 shows the attack results for the UPFD politifact dataset. The results are similar to the ones from the UPFD gossipcop dataset discussed in § 6 and shown in Fig. 5.

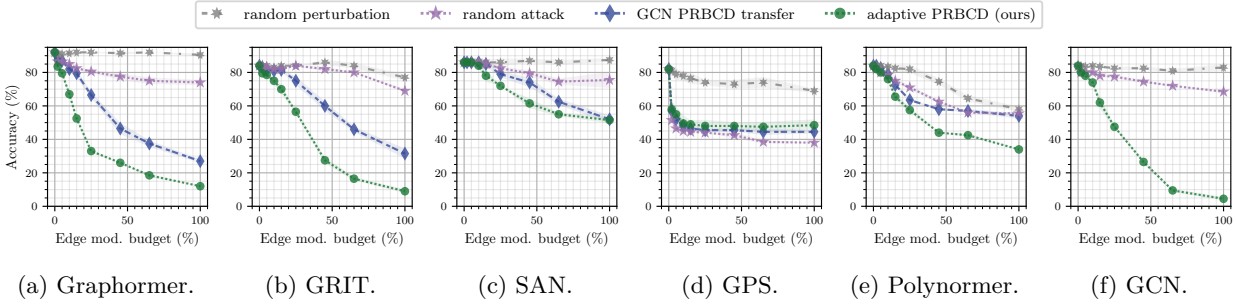

(a) Graphormer.    (b) GRIT.    (c) SAN.    (d) GPS.    (e) Polynormer.    (f) GCN.

Figure 10: Node injection (evasion) attack results for UPFD politifact (graph classification).

## C.3 Best Transfer Attacks

Here we provide the results for the best transfer results, analogous to Fig. 6 but for all additional datasets. Results for CLUSTER are in Fig. 11, for CLUSTER (constrained) in Fig. 12, for Reddit Threads in Fig. 13, and for UPFD politifact in Fig. 14.

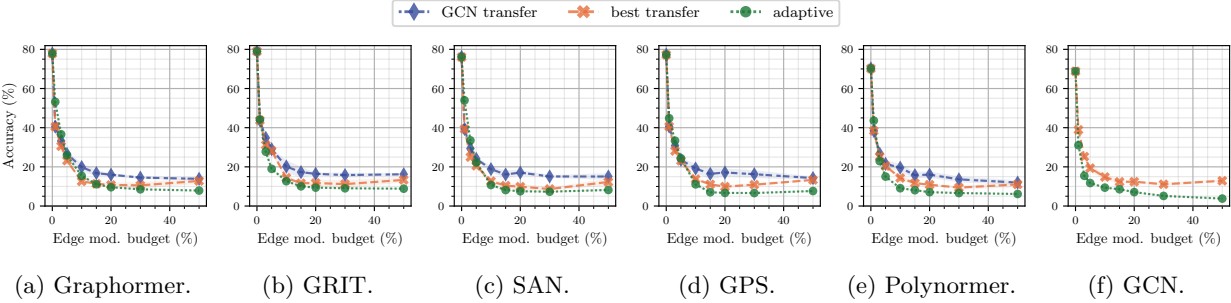

(a) Graphormer.    (b) GRIT.    (c) SAN.    (d) GPS.    (e) Polynormer.    (f) GCN.

Figure 11: Best transfer, CLUSTER (inductive node classification), structure attack (global, evasion).

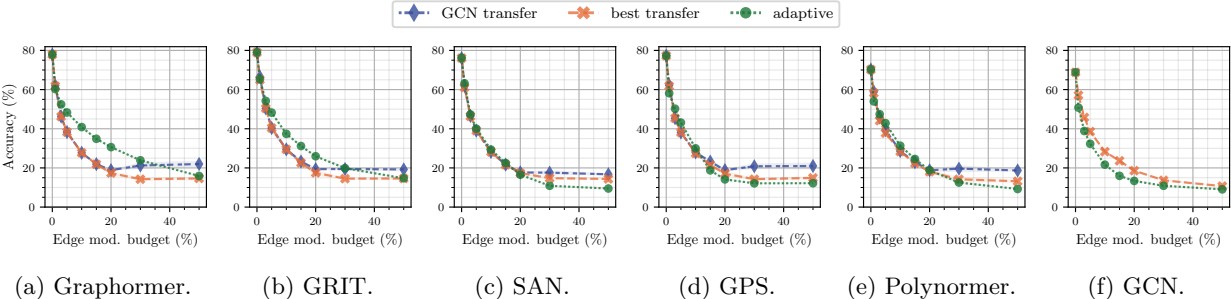

(a) Graphormer.    (b) GRIT.    (c) SAN.    (d) GPS.    (e) Polynormer.    (f) GCN.

Figure 12: Best transfer, CLUSTER (inductive node classification), constrained structure attack (global, evasion).

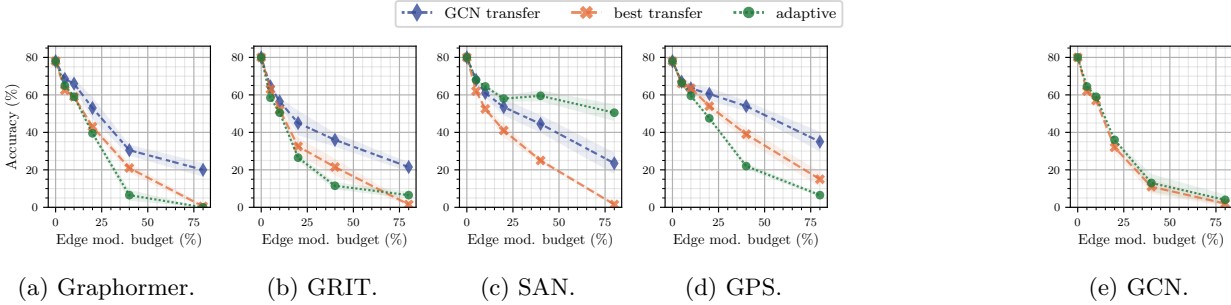

Figure 13: Best transfer, Reddit Threads (graph classification), structure attack (evasion).

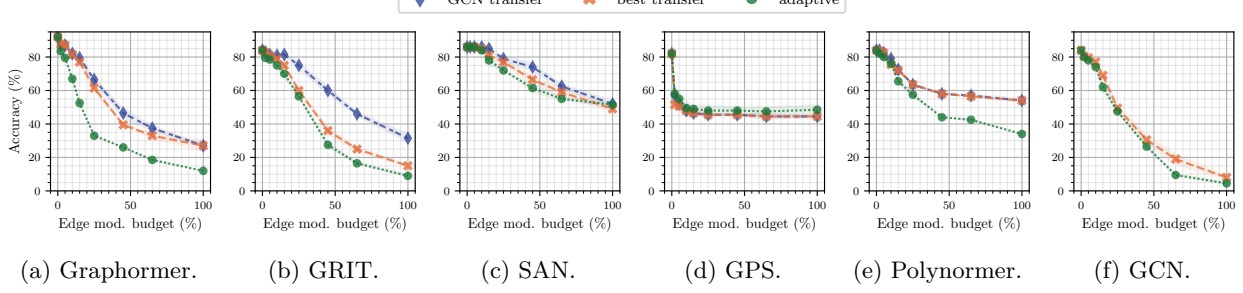

Figure 14: Best transfer, UPFD politifact (graph classification), node injection attack (evasion).

## C.4 All Transfer Results

Here we provide the detailed attack results including the individual transfer models. Results for Graphormer are in Fig. 15, for GRIT in Fig. 16, for SAN in Fig. 17, for GPS in Fig. 18, for Polynormer in Fig. 19, and for GCN in Fig. 20. The results highlight that depending on the dataset, some GT models (e.g., Graphormer and GRIT) transfer well, while other don't (e.g., GPS to Graphormer).

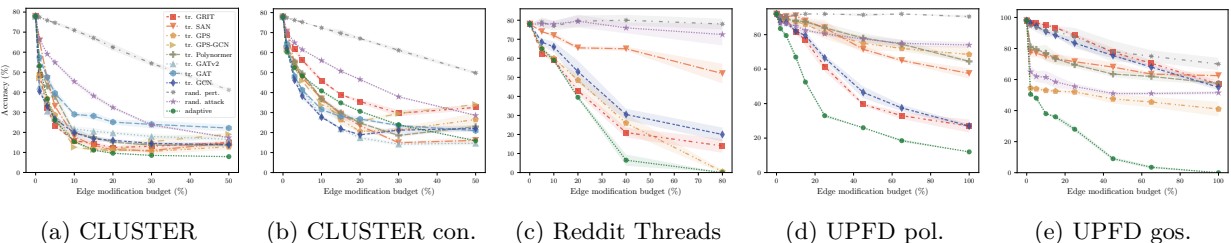

Figure 15: Graphormer attack results with all transfer models shown.

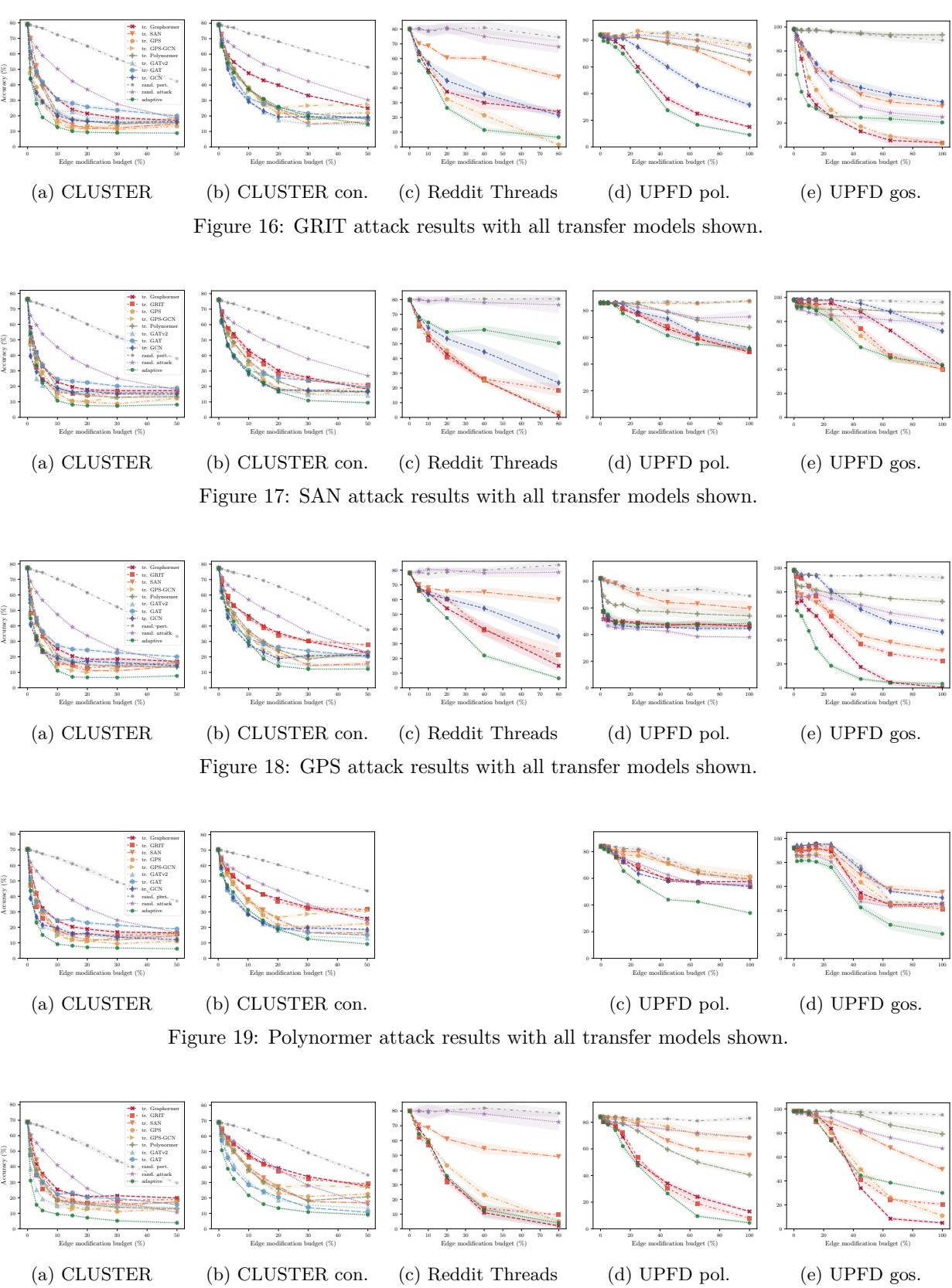

Figure 16: GRIT attack results with all transfer models shown.

Figure 17: SAN attack results with all transfer models shown.

Figure 18: GPS attack results with all transfer models shown.

Figure 19: Polynormer attack results with all transfer models shown.

Figure 20: GCN attack results with all transfer models shown.

### C.5 Ablations

We enable each of the continuous relaxations individually and together in different combinations. We report the results for Graphormer in Tab. 1. The node probability relaxation only applies to the node injection attacks on UPFD. The main insights from the results are: **(a)** All continuous relaxations individually seem to give somewhat useful gradients and can be used to get better results than the gradient-free random baseline. **(b)** For node injection attacks, using only the node probabilities in the graph pooling and to bias the attention scores is usually sufficient and leads to some of the strongest attack results. **(c)** Some relaxations are more effective than others, and using multiple does not seem to always work better than only one. However, one is not consistently better than the other. A good approach might be to try the relaxations individually, to find which are most relevant. Similar effects have been reported by Tramèr et al. (2020) (Recurring Attack Theme T2). We also show ablations for GRIT and SAN components in Tab. 2 and Tab. 3 respectively, from which we can draw the similar conclusions.

We also check the attack strength for GRIT when enabling or disabling gradient computation through certain parts of the model and show the results in Tab. 2. It is possible to get strong attacks even without computing gradients through RRWP, which could be much more efficient computationally, depending on the model and graph size. For node injection attacks, as for the other models, using only the node probability bias in the attention scores already leads to the strongest attacks we report.

Ablation on different attack components on SAN are presented in Tab. 3. The colomn 'Eig. backp.' refers to the alternative method of obtaining gradients through the eigen-decomposition discussed in § F.2. The results indicate that both methods seem to work equally well.

Table 1: Ablations for the Graphormer relaxations for a fixed budget of 1% for CLUSTER without and with perturbation constraints (c.), and 10% for UPFD politifact (pol.) and gossipcop (gos.). The mean and standard deviation over 4 runs with different seeds are reported.

| Deg. | SPD | Acc. (%) | | Node prob. | Acc. (%) | |
|---|---|---|---|---|---|---|
| | | CLUSTER | CLUSTER c. | | UPFD pol. | UPFD gos. |
| ✓ | ✓ | $52.61 \pm 0.57$ | $\mathbf{60.00} \pm 0.42$ | ✓ | $67.0 \pm 2.0$ | $\mathbf{38.0} \pm 0.0$ |
| ✓ | | $\mathbf{46.78} \pm 0.46$ | $68.45 \pm 0.37$ | ✓ | $67.0 \pm 2.0$ | $\mathbf{38.0} \pm 0.0$ |
| | ✓ | $50.81 \pm 0.41$ | $60.66 \pm 0.21$ | ✓ | $\mathbf{66.5} \pm 1.9$ | $39.5 \pm 1.9$ |
| | | | | ✓ | $\mathbf{66.5} \pm 1.9$ | $38.5 \pm 1.0$ |
| ✓ | ✓ | | | | $80.5 \pm 3.4$ | $53.5 \pm 1.0$ |
| random | | $66.52 \pm 0.61$ | $70.29 \pm 0.32$ | | $85.0 \pm 2.6$ | $61.5 \pm 4.1$ |
| clean | | $77.89$ | $77.89$ | | $92.0$ | $98.0$ |

Table 2: Ablations for the GRIT relaxations for a fixed budget of 1% for CLUSTER without and with perturbation constraints (c.), and 10% for UPFD politifact (pol.) and gossipcop (gos.). The mean and standard deviation over 4 runs with different seeds are reported.

| PE grad. | Deg. grad. | Acc. (%) | | Node prob. | Acc. (%) | |
|---|---|---|---|---|---|---|
| | | CLUSTER | CLUSTER c. | | UPFD pol. | UPFD gos. |
| ✓ | ✓ | $\mathbf{44.07} \pm 0.79$ | $\mathbf{65.25} \pm 0.22$ | ✓ | $\mathbf{34.5} \pm 1.0$ | $75.0 \pm 2.6$ |
| ✓ | | $46.27 \pm 0.36$ | $65.70 \pm 0.35$ | ✓ | $\mathbf{34.5} \pm 1.0$ | $74.5 \pm 2.5$ |
| | ✓ | $49.51 \pm 0.90$ | $66.49 \pm 0.49$ | ✓ | $\mathbf{34.5} \pm 1.0$ | $\mathbf{73.5} \pm 1.0$ |
| | | | | ✓ | $\mathbf{34.5} \pm 1.0$ | $\mathbf{73.5} \pm 1.0$ |
| ✓ | ✓ | | | | $54.5 \pm 1.9$ | $83.0 \pm 2.0$ |
| random | | $69.13 \pm 0.10$ | $72.25 \pm 0.29$ | | $76.0 \pm 4.3$ | $82.0 \pm 0.0$ |
| clean | | $78.98$ | $78.98$ | | $98.0$ | $84.0$ |

Table 3: Ablations for the SAN relaxations for a fixed budget of 1% for CLUSTER without and with perturbation constraints (c.), and 10% for UPFD politifact (pol.) and gossipcop (gos.). The mean and standard deviation over 4 runs with different seeds are reported.

| Attn. | Lap. pert. | Eig. backp. | Acc. (%) | | Node prob. | Acc. (%) | |
|---|---|---|---|---|---|---|---|
| | | | CLUSTER | CLUSTER c. | | UPFD pol. | UPFD gos. |
| ✓ | ✓ | | $54.0 \pm 0.6$ | $63.3 \pm 0.3$ | ✓ | $83.5 \pm 1.0$ | $91.5 \pm 4.1$ |
| ✓ | | ✓ | $54.4 \pm 0.6$ | $\mathbf{62.9} \pm 0.2$ | ✓ | $82.0 \pm 2.3$ | $94.0 \pm 3.0$ |
| ✓ | | | $\mathbf{53.9} \pm 0.3$ | $63.2 \pm 0.1$ | ✓ | $77.5 \pm 1.0$ | $91.5 \pm 2.5$ |
| | ✓ | | $57.1 \pm 0.6$ | $67.2 \pm 0.2$ | ✓ | $83.5 \pm 1.0$ | $89.5 \pm 1.9$ |
| | | ✓ | $55.1 \pm 1.0$ | $67.3 \pm 0.3$ | ✓ | $81.0 \pm 1.2$ | $89.5 \pm 3.4$ |
| | | | | | ✓ | $\mathbf{77.0} \pm 1.2$ | $89.5 \pm 3.4$ |
| ✓ | ✓ | | | | | $86.0 \pm 0.0$ | $90.1 \pm 6.0$ |
| ✓ | | ✓ | | | | $86.0 \pm 2.3$ | $91.0 \pm 5.3$ |
| random | | | $65.7 \pm 0.7$ | $68.9 \pm 0.3$ | | $86.0 \pm 0.0$ | $\mathbf{87.5} \pm 1.0$ |
| clean | | | $76.1$ | $76.1$ | | $86.0$ | $98.0$ |

## C.6 Unnoticability Results

Here we report the node-centric homophily results as mentioned by Chen et al. (2022) as a semantic unnoticability criteria for node injection attacks. Following Chen et al. (2022), we use the following node centric homophily measure $h_u$ for a node $u$:

$$h_u = \frac{r_u \cdot X_u}{\|r_u\|_2 \|X_u\|_2}, \ r_u = \sum_{j \in \mathcal{N}(u)} \frac{1}{\sqrt{d_j}\sqrt{d_u}} X_j \tag{28}$$

where $X_u$ is the feature vector of node $u$. As we work with graph classification datasets, we compute $h_u$ for every node in every training graph and of the first 50 test graphs. Figure 21 and Figure 22 show the results for the UPFD gossipcop dataset with an adversarial budget of 5% and 25%, respectively. They highlight that there is no significant homophily change towards lower homophily values w.r.t. the unperturbed test graphs or the original training graphs and thus, establishes that our attacks are unnoticable w.r.t. the metric proposed by Chen et al. (2022). This does not come as a big surprise, because in our node injection attack, we do not treat the node features as part of the optimization problem and only allow connections to nodes in other graphs (for our dataset these are other real users), which potentially more resembles a classic graph modification attack regarding homophily changes (Chen et al., 2022). For Polynormer, one can observe a slight reduction in the highest homophily values for a stronger attack, interestingly, for the other models, we find that higher attack budgets increase the overall measured homophily. We think that a potential explanation lies in the already highly homophilic data. Adding more edges though higher attack budgets leads to even more averaging of already highly homophilic data. This also indicates that the actual predictive feature difference in the dataset as used by the GTs is not capturable by node-centric homophily.

Figure 21 and Figure 22 show similar results for the UPFD politifact dataset with an adversarial budget of 5% and 25%, respectively. Here, we observe the same patterns of no big changes to the homophily. However, the highest homophilic measurements actually slightly decrease as the attack strength increases, whereas close to highest homophilic measurements increase.

Lastly, we want to note that no perturbed test-graph has a lower homophily score than found in the training graphs and thus, the trivial defense against node injection attacks proposed by Chen et al. (2022) based on cutting away all nodes with lower homohpily scores than found in the training data won't be effective.

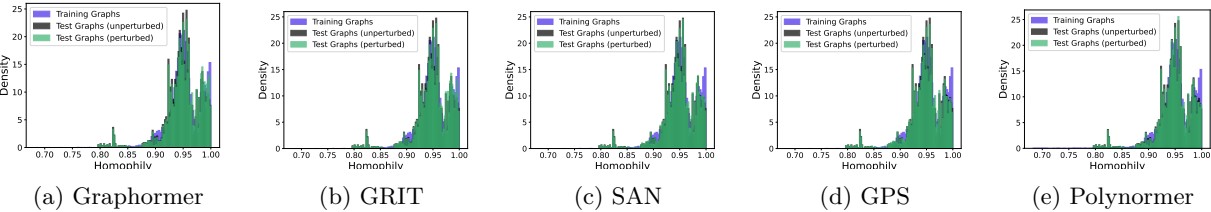

| (a) Graphormer | (b) GRIT | (c) SAN | (d) GPS | (e) Polynormer |

Figure 21: Node-centric homophilies in the training set, unperturbed test set and perturbed test set for $\epsilon = 0.05$ in the UPFD gossipcop dataset.

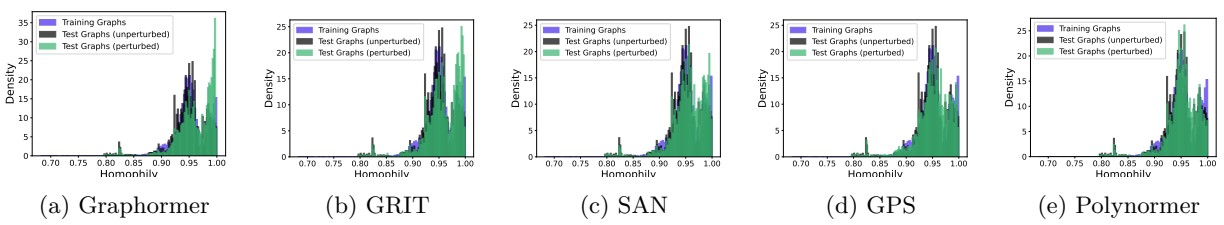

| (a) Graphormer | (b) GRIT | (c) SAN | (d) GPS | (e) Polynormer |

Figure 22: Node-centric homophilies in the training set, unperturbed test set and perturbed test set for $\epsilon = 0.25$ in the UPFD gossipcop dataset.

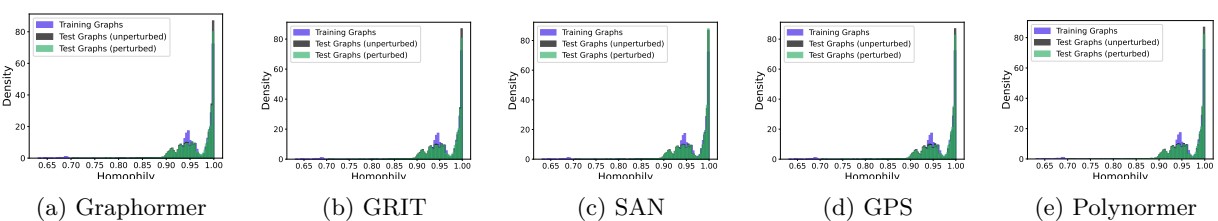

| (a) Graphormer | (b) GRIT | (c) SAN | (d) GPS | (e) Polynormer |

Figure 23: Node-centric homophilies in the training set, unperturbed test set and perturbed test set for $\epsilon = 0.05$ in the UPFD politifact dataset.

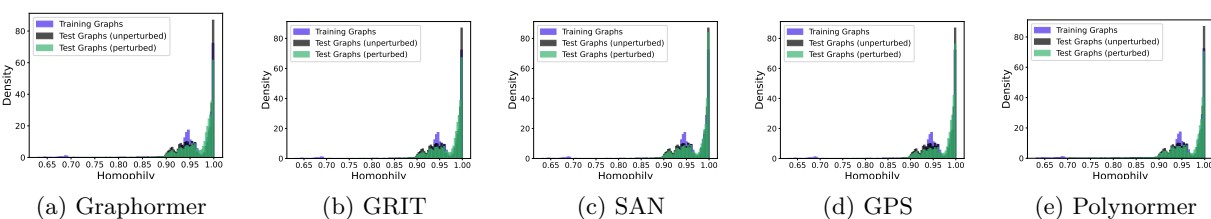

| (a) Graphormer | (b) GRIT | (c) SAN | (d) GPS | (e) Polynormer |

Figure 24: Node-centric homophilies in the training set, unperturbed test set and perturbed test set for $\epsilon = 0.25$ in the UPFD politifact dataset.

## C.7 Local Attack Results

Figures 25 and 26 show the results of applying our adaptive attacks in a local attack setting for CLUSTER. In Figure 25, we use the constrained setting (see Section 6) that prohibits modifying edges to the labeled nodes, to increase the difficulty of the attack setting. The results highlight the efficacy of our adaptive attacks, significantly outperforming the random-attack baseline. Importantly, our adaptive attacks show similar strength w.r.t. the random baseline as PR-BCD does on GCN, which is known to be stronger then Nettack (Geisler et al., 2021; Zügner et al., 2018) in the local attack setting.

To allow comparison with Nettack (Zügner et al., 2018), we explore another attack setting, as its threat model definition does not allow to model the above constrained setting. In particular, Nettack defines a set of attack nodes $\mathcal{A}$ and allows the deletion or addition of an edge $(u, v)$ if either $u \in \mathcal{A}$, or $v \in \mathcal{A}$. Now, to compare Nettack with our adaptive attacks, we set $\mathcal{A}$ to the 1-hop neighborhood $\mathcal{N}(u)$ of the given target node $u$, and constrain our adaptive attacks to the same set of adversarial edge modifications. We say an attack budget is small to moderate if it can perturb at most $\frac{|N(u)|}{2}$ edges and is large if it can perturb between $\frac{|N(u)|}{2}$ to $|N(u)|$ edges. Even stronger attacks have high risks of semantic violations (Gosch et al., 2023b). On average on the CLUSTER dataset, the attack budgets we consider of 0.2% and 0.5% correspond to small to moderate budgets (4 and 10 perturbations), 0.9% and 1.4% to strong budgets (18 and 28 perturbations) and 2% to 40 perturbations. Figure 26 highlights that for small to moderate budgets, our adaptive attacks outperform Nettack by large margins. This highlights the effectiveness of the gradient signal derived from our relaxations. For large budgets, our adaptive attacks outperform Nettack for GRIT and GPS, while Nettack gets strong results for SAN and Polynormer. Note here that the transferability between attacks on CLUSTER is in general high (see Appendix C.4), which benefits transfer attacks like Nettack.

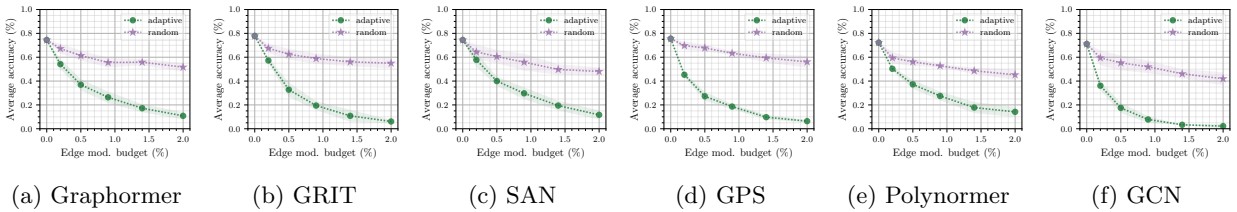

| (a) Graphormer | (b) GRIT | (c) SAN | (d) GPS | (e) Polynormer | (f) GCN |

Figure 25: Local adaptive attacks on CLUSTER (constrained setting, see Section 6). Edge modification budgets are shown from 0% to 2%.

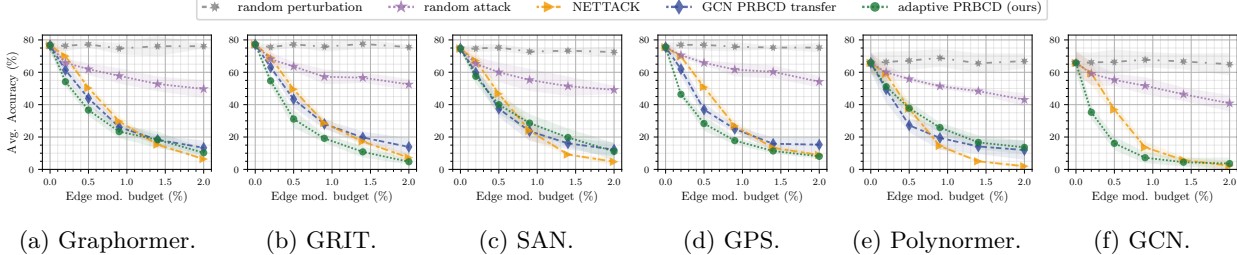

| (a) Graphormer. | (b) GRIT. | (c) SAN. | (d) GPS. | (e) Polynormer. | (f) GCN. |

Figure 26: Local adaptive attacks on CLUSTER (1-hop neighborhood). Edge modification budgets 0.2% and 0.5% on average correspond to small to medium attack budgets (i.e., the number of edges modifications is $\leq$ than half of the victim node's neighborhood size) with our adaptive attack consistently outperforming Nettack for these budgets. 0.9% and 1.4% usually correspond to strong attack budgets (between half of the nodes neighborhood size to the node's neighborhood size). 2% allows to perturb more edges than the size of the victim node's neighborhood.

# D   Adversarial Training Algorithm

We based our implementation of the adversarial training on the 'Free' adversarial training of Shafahi et al. (2019). Pseudocode for our adversarial training is given in Alg. 1. The main modifications are that we do two separate forward and backwards passes for the attack and model respectively. This is because: (1) the attack and model often have distinct loss functions that they optimize for, and (2) we sample a discrete structure perturbation for the model, such that the perturbed graph is included in the original valid sample space. Another difference is that we need to iterate over the graphs in the minibatch separately. This is a limitation caused by: (1) The attack optimization steps are not trivial to parallelize, especially for our node injection attacks, and (2) the PE computations (e.g. Laplacian eigen-decomposition) are also not easily parallelizable and need to re-computed for each new perturbed graph.

Given these limitations, our adversarial training is much less efficient. It requires at least $2 \cdot |\mathcal{B}|$ times more model evaluations than normal training. Furthermore, for many GTs the PE computation is one of the most computationally expensive steps. Therefore, PEs are usually precomputed in a pre-processing step. During adversarial training, we need to compute PEs for new unseen perturbations at each step, which further increases the overhead. Nonetheless, following the main idea of Shafahi et al. (2019) alleviates some of the overhead and makes it somewhat practically feasible.

---

**Algorithm 1** Our $k$-step 'free' adversarial training

---

**Require:** Training dataset $\mathcal{T}$, model $f_\theta$, attack budget $\Delta$, number of steps $k$, learning rate $\alpha$

  Initialize $\theta$
  **for** epoch $= 1...N_{ep}/k$ **do**
    **for** minibatch $\mathcal{B} \subset \mathcal{T}$ **do**
      Initialize perturbations $\boldsymbol{P}$
      **for** $i = 1...k$ **do**
        $g_\theta \leftarrow 0$
        **for** graph $\mathcal{G} = (\boldsymbol{A}, \boldsymbol{X}, \boldsymbol{y}) \in \mathcal{B}$ **do**
          $\boldsymbol{P} \leftarrow \text{PRBCD\_step}(f_\theta, \boldsymbol{X}, \boldsymbol{A}, \boldsymbol{P}, \Delta)$
          $\boldsymbol{A}' \leftarrow \text{sample\_discrete}(\boldsymbol{A}, \boldsymbol{P})$
          $g_\theta \leftarrow g_\theta \nabla_\theta \mathcal{L}(f_\theta(\boldsymbol{A}', \boldsymbol{X}), \boldsymbol{y})$
        **end for**
        $\theta \leftarrow \theta + \alpha \cdot \frac{1}{|\mathcal{B}|} \cdot g_\theta$
      **end for**
    **end for**
  **end for**

---

# E   Hyperparameters

To obtain trained models of comparable performance for each architecture type, we performed a hyperparameter search for each model and dataset. Because of the large number of experiments and hyperparameters we used random sampling of hyperparameter values in predetermined ranges. These ranges are defined and available in the configuration files in our code repository under `https://github.com/isefos/gt_robustness`. The final hyperparameters of the best models used for the robustness results are shown for Graphormer in Tab. 4, for SAN in Tab. 5, for GRIT in Tab. 6, for Polynormer in Tab. 8, for GPS in Tab. 7, for GCN in Tab. 9, for GPS-GCN in Tab. 10, for GAT in Tab. 11, and for GATv2 in Tab. 12.

Table 4: Hyperparameters for Graphormer.

|  | CLUSTER | Reddit Threads | UPFD gos. | UPFD pol. |
|---|---|---|---|---|
| Optimizer | adam | adamW | adamW | adamW |
| Learning rate | $8.04 \times 10^{-4}$ | $1.05 \times 10^{-4}$ | $3.75 \times 10^{-4}$ | $1.29 \times 10^{-4}$ |
| Weight decay | 0 | $1.18 \times 10^{-6}$ | $1.37 \times 10^{-5}$ | $6.7 \times 10^{-3}$ |
| PE max. degree | 70 | 42 | 21 | 31 |
| SPD max. distance | 4 | 8 | 8 | 10 |
| Attention dropout | 0.107 | 0.138 | 0.382 | 0 |
| Input dropout | 0 | 0 | $8.65 \times 10^{-3}$ | 0 |
| MLP dropout | $4.47 \times 10^{-2}$ | $1.21 \times 10^{-2}$ | $5.37 \times 10^{-2}$ | 0 |
| Hidden dimension | 60 | 48 | 30 | 40 |
| Layers | 15 | 6 | 8 | 6 |
| Attention heads | 6 | 8 | 3 | 8 |
| Graph pooling | - | virtual node | virtual node | virtual node |

Table 5: Hyperparameters for SAN.

|  | CLUSTER | Reddit Threads | UPFD gos. | UPFD pol. |
|---|---|---|---|---|
| Optimizer | adam | adamW | adam | adam |
| Learning rate | $5.0 \times 10^{-4}$ | $5.66 \times 10^{-4}$ | $5.41 \times 10^{-4}$ | $1.29 \times 10^{-4}$ |
| Weight decay | 0 | $3.54 \times 10^{-7}$ | 0 | $1.0 \times 10^{-3}$ |
| $k$ (num. eig.) | 10 | 18 | 24 | 10 |
| PE dimension | 16 | 8 | 20 | 16 |
| PE layers | 1 | 1 | 2 | 2 |
| PE heads | 4 | 8 | 5 | 4 |
| $\gamma$ (global/ local) | 0.1 | $5.46 \times 10^{-5}$ | $4.28 \times 10^{-3}$ | $1.43 \times 10^{-2}$ |
| Dropout | 0 | $7.92 \times 10^{-2}$ | $1.73 \times 10^{-2}$ | 0 |
| Hidden dimensions | 48 | 48 | 80 | 96 |
| Layers | 16 | 7 | 3 | 3 |
| Attention heads | 8 | 8 | 8 | 4 |
| Graph pooling | - | add | mean | add |

Table 6: Hyperparameters for GRIT.

|  | CLUSTER | Reddit Threads | UPFD gos. | UPFD pol. |
|---|---|---|---|---|
| Optimizer | adamW | adamW | adamW | adamW |
| Learning rate | $1.29 \times 10^{-3}$ | $8.02 \times 10^{-4}$ | $2.24 \times 10^{-3}$ | $5.61 \times 10^{-4}$ |
| Weight decay | $4.16 \times 10^{-6}$ | $3.05 \times 10^{-8}$ | $1.2 \times 10^{-8}$ | $2.97 \times 10^{-2}$ |
| RRWP max. steps | 4 | 6 | 6 | 9 |
| Attention dropout | 0.478 | 0.28 | 0.293 | 0.49 |
| Dropout | $1.0 \times 10^{-2}$ | $1.05 \times 10^{-2}$ | $5.55 \times 10^{-2}$ | 0 |
| Hidden dimensions | 48 | 24 | 18 | 9 |
| Layers | 12 | 11 | 6 | 2 |
| Attention heads | 8 | 8 | 6 | 3 |
| Graph pooling | - | mean | add | mean |

Table 7: Hyperparameters for GPS.

|  | CLUSTER | Reddit Threads | UPFD gos. | UPFD pol. |
|---|---|---|---|---|
| Optimizer | adamW | adamW | adamW | adamW |
| Learning rate | $5.0 \times 10^{-4}$ | $3.21 \times 10^{-3}$ | $4.28 \times 10^{-4}$ | $1.18 \times 10^{-2}$ |
| Weight decay | $1.0 \times 10^{-5}$ | $7.44 \times 10^{-3}$ | $6.34 \times 10^{-8}$ | $1.3 \times 10^{-4}$ |
| $k$ (num. eig.) | 10 | 13 | 10 | 10 |
| PE dimension | 16 | 16 | 16 | 16 |
| PE encoder | DeepSet | DeepSet | DeepSet | DeepSet |
| Attention dropout | 0.1 | $9.32 \times 10^{-2}$ | 0.1 | 0.1 |
| Dropout | 0.1 | $9.32 \times 10^{-2}$ | 0.1 | 0.1 |
| Hidden dimensions | 48 | 24 | 40 | 32 |
| Layers | 16 | 7 | 5 | 6 |
| Attention heads | 8 | 8 | 8 | 8 |
| Graph pooling | - | add | add | add |

Table 8: Hyperparameters for Polynormer.

|  | CLUSTER | UPFD gos. | UPFD pol. |
|---|---|---|---|
| Optimizer | adamW | adamW | adamW |
| Learning rate | $2.35 \times 10^{-3}$ | $8.93 \times 10^{-4}$ | $1.72 \times 10^{-4}$ |
| Weight decay | $1.0 \times 10^{-7}$ | $1.0 \times 10^{-7}$ | $1.0 \times 10^{-7}$ |
| Attention dropout | $5.0 \times 10^{-2}$ | $5.0 \times 10^{-2}$ | $5.0 \times 10^{-2}$ |
| Dropout local | 0.137 | 0.149 | 0.121 |
| Dropout global | $5.0 \times 10^{-2}$ | $5.0 \times 10^{-2}$ | $5.0 \times 10^{-2}$ |
| Hidden dimensions | 72 | 48 | 32 |
| Layers local | 16 | 13 | 3 |
| Layers global | 3 | 4 | 1 |
| Attention heads local | 8 | 8 | 8 |
| Attention heads global | 8 | 8 | 8 |
| Graph pooling | - | add | add |

Table 9: Hyperparameters for GCN.

|  | CLUSTER | Reddit Threads | UPFD gos. | UPFD pol. |
|---|---|---|---|---|
| Optimizer | adam | adamW | adamW | adamW |
| Learning rate | $1.0 \times 10^{-3}$ | $3.17 \times 10^{-3}$ | $1.23 \times 10^{-4}$ | $5.29 \times 10^{-3}$ |
| Weight decay | 0 | $2.7 \times 10^{-6}$ | $2.85 \times 10^{-6}$ | $2.59 \times 10^{-2}$ |
| Dropout | 0 | $3.82 \times 10^{-3}$ | 0.5 | 0 |
| Hidden dimension | 172 | 30 | 105 | 473 |
| Layers | 16 | 8 | 3 | 2 |
| Graph pooling | - | mean | add | mean |

Table 10: Hyperparameters for GPS-GCN.

|  | CLUSTER |
|---|---|
| Optimizer | adamW |
| Learning rate | $9.74 \times 10^{-4}$ |
| Weight decay | $1.0 \times 10^{-8}$ |
| $k$ (num. eig.) | 10 |
| PE dimension | 16 |
| PE encoder | DeepSet |
| Attention dropout | $5.0 \times 10^{-2}$ |
| Dropout | $5.0 \times 10^{-2}$ |
| Hidden dimensions | 40 |
| Layers | 13 |
| Attention heads | 8 |
| Graph pooling | - |

Table 11: Hyperparameters for GAT.

|  | CLUSTER |
|---|---|
| Optimizer | adam |
| Learning rate | $1.0 \times 10^{-3}$ |
| Weight decay | 0 |
| Dropout | 0 |
| Hidden dimension | 176 |
| Layers | 16 |
| Attention heads | 8 |
| Graph pooling | - |

Table 12: Hyperparameters for GATv2.

|  | CLUSTER |
|---|---|
| Optimizer | adam |
| Learning rate | $1.0 \times 10^{-3}$ |
| Weight decay | 0 |
| Dropout | 0 |
| Hidden dimension | 120 |
| Layers | 16 |
| Attention heads | 8 |
| Graph pooling | - |

# F   Laplacian Eigen-Decomposition Gradient

## F.1   Perturbation Approximation: Repeated Eigenvalues

Unfortunately, Eq. 18 and 19 do not hold in general when repeated eigenvalues are present. This is due to the fact that a small perturbation can separate repeated eigenvalues into distinct eigenvalues. For the unperturbed graph, the choice of eigenvector basis of the repeated eigenvalue's eigenspace is arbitrary. In the perturbed graph, however, the eigenvectors corresponding to the now distinct eigenvalues are uniquely defined (up to the sign). Thus, a large discontinuous change in the eigenvectors can be caused by an arbitrarily small input perturbation. For instance, consider the matrix $\boldsymbol{M}$ with repeated eigenvalue 1 and the following valid eigendecomposition:

$$\boldsymbol{M} = \begin{bmatrix} 1 & 0 \\ 0 & 1 \end{bmatrix} = \boldsymbol{U}\boldsymbol{\Lambda}\boldsymbol{U}^{\mathsf{T}}, \quad \boldsymbol{\Lambda} = \begin{bmatrix} 1 & 0 \\ 0 & 1 \end{bmatrix}, \quad \boldsymbol{U} = \frac{\sqrt{2}}{2}\begin{bmatrix} 1 & 1 \\ 1 & -1 \end{bmatrix} \tag{29}$$

As soon as an arbitrarily small perturbation $\varepsilon$ is added to one of the diagonal entries, the eigenvalues become distinct and the choice of eigenvectors becomes constrained, which results in a discontinuous change:

$$\tilde{\boldsymbol{M}} = \begin{bmatrix} 1 & 0 \\ 0 & 1+\varepsilon \end{bmatrix} = \tilde{\boldsymbol{U}}\tilde{\boldsymbol{\Lambda}}\tilde{\boldsymbol{U}}^{\mathsf{T}}, \quad \tilde{\boldsymbol{\Lambda}} = \begin{bmatrix} 1 & 0 \\ 0 & 1+\varepsilon \end{bmatrix}, \quad \tilde{\boldsymbol{U}} = \begin{bmatrix} 1 & 0 \\ 0 & 1 \end{bmatrix} \tag{30}$$

However, there is always some valid choice of eigenvectors in the unperturbed graph that leads to a continuous change with respect to the given perturbation, e.g., in the above example $\tilde{\boldsymbol{U}}$ is also a valid choice for the eigenvectors $\boldsymbol{U}$ of the unperturbed matrix. With the right choice of unperturbed eigenvectors, the approximation equations are, therefore, still valid. Here, we provide a procedure to transform arbitrary eigenvectors into the ones that lead to good perturbation approximations. For the theory showing why this leads to the correct result, we refer to Bamieh (2022).

Let $(\boldsymbol{\Lambda}, \hat{\boldsymbol{U}})$ be the output of the eigendecomposition algorithm for the unperturbed Laplacian $\boldsymbol{L}_{sym}$ containing repeated eigenvalues. We can write the eigendecomposition in it's block form:

$$\boldsymbol{L}_{sym} = \hat{\boldsymbol{U}}\boldsymbol{\Lambda}\hat{\boldsymbol{U}}^{\mathsf{T}}, \quad \boldsymbol{\Lambda} = \begin{bmatrix} \boldsymbol{\Lambda}_1 & & \\ & \ddots & \\ & & \boldsymbol{\Lambda}_{n'} \end{bmatrix}, \quad \hat{\boldsymbol{U}} = \begin{bmatrix} | & & | \\ \hat{\boldsymbol{U}}_1 & \cdots & \hat{\boldsymbol{U}}_{n'} \\ | & & | \end{bmatrix} \tag{31}$$

For a simple eigenvalue $\lambda_i$, the block has dimension one, i.e., $\boldsymbol{\Lambda}_i = [\lambda_i]$ and $\hat{\boldsymbol{U}}_i = \boldsymbol{u}_i$. For a repeated eigenvalue $\lambda_j$ with multiplicity $r$, it's corresponding block is $\lambda_j\boldsymbol{I}_r$ and $\hat{\boldsymbol{U}}_j \in \mathbb{R}^{n \times r}$. Let $\boldsymbol{P} = \boldsymbol{P}^{\mathsf{T}}$ be an arbitrary symmetric perturbation to the original symmetric Laplacian. We can transform each eigenspace basis of a repeated eigenvalue $\hat{\boldsymbol{U}}_j$ to the correct choice of eigenvectors as follows:

$$\begin{aligned} \boldsymbol{U}_j &= \hat{\boldsymbol{U}}_j\boldsymbol{Q} \\ \boldsymbol{P}_{\hat{U},j} &= \hat{\boldsymbol{U}}_j^{\mathsf{T}}\boldsymbol{P}\hat{\boldsymbol{U}}_j = \boldsymbol{Q}\boldsymbol{\Lambda}_P\boldsymbol{Q}^{\mathsf{T}} \ \in \mathbb{R}^{r \times r} \end{aligned} \tag{32}$$

First, we do a basis transformation of the perturbation matrix onto the eigenbasis $\hat{\boldsymbol{U}}$. Then we find the eigendecomposition of the corresponding diagonal block $\boldsymbol{P}_{\hat{U},j}$ and use these perturbation eigenvectors to transform the original Laplacian eigenvectors. This results in a choice of valid eigenvectors $\boldsymbol{U}_j$ such that the approximations in Eq. 18 and 19 are valid for repeated eigenvalues and guarantees continuity of the eigenvalues and vectors with respect to a single perturbation, e.g., when linearly interpolating from the unperturbed to the fully perturbed matrix.

## F.2   Backpropagation: Breaking Up Repeated Eigenvalues

The only thing preventing the use of auto-differentiation to compute gradients through the eigen-decomposition is the presence of repeated eigenvalues. As a workaround, Lin et al. (2022) propose adding small amplitude random noise to the entire adjacency matrix. While this usually separates the repeated eigenvalues, it is

not guaranteed to. We propose a different approach in which the smallest possible perturbation term is added to the Laplacian matrix, such that the repeated eigenvalues are guaranteed to be separated while the eigenvectors remain unchanged.

To achieve this, we must first define a minimum eigenvalue distance hyperparameter $\varepsilon$, which we set to $10^{-4}$ in our experiments. Then we define eigenvalue separation such that for all perturbed Laplacian eigenvalues $|\hat{\lambda}_i - \hat{\lambda}_j| \geq \varepsilon$ must hold. Furthermore, we can define a vector $\boldsymbol{o} \in \mathbb{R}^n$ such that each entry represents the offset of the perturbed eigenvalue in relation to the true value:

$$\hat{\boldsymbol{\Lambda}} = \begin{bmatrix} \lambda_1 + \boldsymbol{o}_1 & & \\ & \ddots & \\ & & \lambda_n + \boldsymbol{o}_n \end{bmatrix} = \boldsymbol{\Lambda} + \mathsf{diag}(\boldsymbol{o}) \tag{33}$$

In order for the perturbed matrix to have the same eigenvectors as the unperturbed Laplacian, we can define it by its eigendecomposition:

$$\begin{aligned} \hat{\boldsymbol{L}}_{sym} &= \boldsymbol{U}\hat{\boldsymbol{\Lambda}}\boldsymbol{U}^\mathsf{T} \\ &= \boldsymbol{U}(\boldsymbol{\Lambda} + \mathsf{diag}(\boldsymbol{o}))\boldsymbol{U}^\mathsf{T} \\ &= \boldsymbol{U}\boldsymbol{\Lambda}\boldsymbol{U}^\mathsf{T} + \boldsymbol{U}\mathsf{diag}(\boldsymbol{o})\boldsymbol{U}^\mathsf{T} \\ &= \boldsymbol{L}_{sym} + \boldsymbol{U}\mathsf{diag}(\boldsymbol{o})\boldsymbol{U}^\mathsf{T} \end{aligned} \tag{34}$$

Consequently, the additive perturbation has the form $\boldsymbol{P} = \boldsymbol{U}\mathsf{diag}(\boldsymbol{o})\boldsymbol{U}^\mathsf{T}$, such that it shares the same eigenvectors as the original Laplacian, and its eigenvalues are exactly the offsets.

Since the Frobenius norm can also be computed using the singular values, finding the perturbation with minimum norm is equivalent to minimizing the Euclidean norm of the offset vector $\|\boldsymbol{P}\|_F = \sqrt{\sum_i \boldsymbol{o}_i^2} = \|\boldsymbol{o}\|_2$. To ensure that the order of the eigenvalues is not changed we can define the separation constraints for the consecutive pairs of the perturbed eigenvalues $\hat{\lambda}_{i+1} - \hat{\lambda}_i = (\lambda_{i+1} + \boldsymbol{o}_{i+1}) - (\lambda_i - \boldsymbol{o}_i) \geq \varepsilon$. The total constrained optimization problem can be written as:

$$\begin{aligned} \min_{\boldsymbol{o}} \quad & \frac{1}{2}\|\boldsymbol{o}\|_2^2 \\ \text{subject to} \quad & \boldsymbol{o}_{i+1} - \boldsymbol{o}_i \geq \varepsilon - (\lambda_{i+1} - \lambda_i) \end{aligned} \tag{35}$$

The $(n-1)$ inequality constraints are linear and can be written in matrix-vector form. To further ensure that the total range of the eigenvalues is not changed, the equality constraints $\boldsymbol{o}_0 = \boldsymbol{o}_n = 0$ can be added. As an initial guess, the offsets can be set to equally separate the eigenvalues in their range, which is guaranteed to satisfy all constraints. The optimal solution $\boldsymbol{o}^*$ can be calculated efficiently using constrained optimization.

In conclusion, using the slightly perturbed Laplacian $\hat{\boldsymbol{L}}_{sym} = \boldsymbol{L}_{sym} + \boldsymbol{U}\mathsf{diag}(\boldsymbol{o}^*)\boldsymbol{U}^\mathsf{T}$ as input to the eigendecomposition in the forward pass results in usable gradient via back-propagation. Note that to get the perturbation, the eigendecomposition of the original Laplacian has to be computed. Thus, it can be checked for the presence of repeated eigenvalues, and a second perturbed eigendecomposition is only computed when necessary. Tab. 3 includes results using this approach, which seems to work about as well as the perturbation approximation.

## G  Further Discussions

### G.1  On Principle II

As discussed in Section 3, Principle II requires of $\tilde{f}_\theta$ (i) continuity w.r.t. $\tilde{\boldsymbol{A}}'$, and (ii) differentiability almost everywhere w.r.t. $\tilde{\boldsymbol{A}}'$. This principle can be better understood by using the ReLU function as an analogy. The ReLU function satisfies both (i) and (ii), but is not continuously differentiable - still, neural networks employing ReLU can be effectively optimized (i.e., trained) using gradient descent. This is possible as for any function satisfying (i) and (ii), the gradient can be computed for all inputs except on a subset having

measure zero, which for ReLU is $\{0\}$. In practice, the non-differentiable points are rarely reached, and if so, due to the continuity, an informative gradient can be defined at every non-differentiable point $x_0$ in a principled manner, by choosing it to be either the left-sided limit $a = \lim_{x \to x_0^-} \frac{f(x)-f(x_0)}{x-x_0}$ or the right-sided limit $b = \lim_{x \to x_0^+} \frac{f(x)-f(x_0)}{x-x_0}$ of the function. Thus, the non-differentiability is not interfering with effective continuous optimization through methods like gradient descent. For ReLU, the left-sided limit is 0 and the right-sided limit is 1, and PyTorch 2.7.1 chooses to return the left-sided limit in case of an input $0$[1].

Furthermore, we do require differentiability and not only continuity, as otherwise, gradient-free (black-box) optimization methods have to be used to solve Equation (1), which is a complex and NP-hard combinatorial optimization problem. In fact, the literature on adversarial attacks for GNNs has converged to showing that the most effective attacks against GNNs are usually gradient-based attacks (Xu et al., 2019; Geisler et al., 2021; Gosch et al., 2023a), which is consistent with the image domain (Tramèr et al., 2020).

### G.2 Numerical Considerations on Relaxed Sparse Attention

The computation of $\tilde{\alpha}_{ij} = \mathsf{softmax}(\tilde{w}_{ij})$ with $\tilde{w}_{ij} = w_{ij} + \log(p_{ij})$ itself is numerically stable, even if $p_{ij} = 0$ for some $i, j \in \mathcal{V}$. However, to ensure that the gradient-taking w.r.t. $\tilde{A}'_{ij}$ defining the "soft attention mask" $\log p_{ij}$ is also numerically well-behaved, we set the minimum value of $p_{ij}$ to some small $\epsilon > 0$, which is a hyperparameter that we choose to be $10^{-9}$ for our experiments. Hence, $p_{ij} = 0$ implying $\tilde{w}_{ij} = -\infty$ is never reached. This follows the strategy employed by the attack framework PR-BCD, which we leverage to execute our attacks (see Section 5). For scalability, PR-BCD iteratively samples a batch of edges and only computes the gradient w.r.t. these edges. It always sets the minimum value of a sampled edge to some small $\epsilon$, to ensure proper gradients can be calculated, even if the edge does not originally exist in the graph.

### G.3 On the Computational Complexity of the Derived Relaxations

In this section, we discuss the computations of the different GT models that dominate their runtime complexity and how our relaxations affect/change these.

**Graphormer.** Depending on the concrete implementation and graph characteristics (sparsity), the most dominant computational step w.r.t. the number $n$ of nodes in the graph in *Graphormer* is the shortest path calculation that has to be done for each pair of nodes, or the full-attention calculation also done between each node. The full node attention calculation is untouched by our relaxation. As the relaxation introduces continuous edge weights and thereby, during the attack iterations, reduces the sparsity in the graph, this can slightly increase the time spent for the shortest path calculation, if a shortest path implementation is chosen that particularly leverages the sparsity in the graph (Cormen et al., 2009). However, this does not change the asymptotic time complexity of the shortest path algorithm. As the relaxation otherwise just introduces a linear interpolation of the PE vectors and the learnable scalars associated to discrete shortest-path distances, which can be computed in constant time, it does not change the asymptotic runtime of Graphormer. Thus we can conclude that our relaxation fulfills Principle III.

**SAN.** SAN has two computations that have the potential to dominate its runtime complexity. First, the Laplacian PEs require to calculate the eigendecomposition of the Laplacian which can have a cubic worst-case complexity in the number of nodes. However, only the lowest $k$ eigenvalues and associated eigenvectors are used where $k$ is a hyperparameter and is usually set to a small value between 10 - 20 (see Table 5), which allows for faster computation. Our relaxation performs a first-order approximation of the eigenvalues and eigenvectors of the perturbed Laplacian (see Equations (18) and (19)). Computing these only requires access to the original eigendecomposition and thus, does not increase the runtime complexity. In the special case of repeated eigenvalues (see Appendix F.1), another eigendecomposition has to be computed, but only of the subblock associated to the repeated eigenvalues. Thus, in the theoretic worst case of all eigenvalues being repeated, two eigendecompositions instead of one have to be computed. However, this does not change the asymptotic runtime complexity and in practice, repeated eigenvalues are a rare event occurring seldom if at all in a graph and thus, has little practical relevance for the runtime. Lastly, SAN employs full node attention through two sparse attention schemes (see Equation (9)), one computed over the neighborhood of a node and

---

[1]As the ReLU function is also convex, this corresponds to a subderivative of the ReLU function

one computed over all non-neighbours and thus, has quadratic complexity w.r.t. the number of nodes. Our relaxation technically relaxes both sparse attention mechanisms to full attention (see Equation (17)). However, this does not change the asymptotic complexity, but can have a slight effect on the speed of computing the attention in practice. Conceptually, it is still a sparse attention mechanism, but due to the continuity of the adjacency matrix paired with gradient-taking using PR-BCD, more edges (but at most $b$, where $b$ is the block-size Geisler et al. (2021)) have non-zero values in practice. To conclude, as the asymptotic runtime of SAN doesn't changed and the relaxation introduces some but not significant additional computation, we can conclude that Principle III is satisfied.

**GRIT.** As for GRIT the relaxation only makes the adjacency matrix continuous without introducing any other changes and a continuous adjacency matrix does not change any of the involved computations for GRIT, its runtime complexity is not affected. Thus, Prinple III is satisfied.

**GPS.** GPS again has two potential dominating computations. First, it uses Laplacian PEs. However, as discussed for SAN, our relaxation does not affect its runtime complexity. The second is its global attention module that is not affected by our relaxation and which dominates the relaxed local GatedGCN computation. Thus, the relaxed GPS also fulfills Principle III.

**Polynormer.** The only change in computation our relaxation introduces is in the sparse attention computation of GAT, by adding $\log p_{ij}$ to the attention logit computation $w_{ij}$. Theoretically, this does not change the asymptotic runtime complexity of the attention computation, because if $p_{ij} = 0$, i.e., there is no edge $(i, j)$, there is no contribution of $w_{ij}$ (i.e., node $j$), to the attention score $\alpha_{ij}$. I.e., the difference is that the neighbourhood of the attention is now defined as $\tilde{A}_{ij} > 0$ instead of $\tilde{A}_{ij} = 1$ in the non-relaxed case. Due to the continuity of the adjacency matrix paired with gradient-taking using PR-BCD, more edges but at most $b$, where $b$ is the block-size (Geisler et al., 2021), have non-zero values in practice. Thus, this can have an effect, though not excessive, on the practical runtime of computing the sparse attention. In our implementation, we choose to follow the presentation of SAN's relaxed spare attention and technically implemented the relaxation as full attention. This can potentially dominate Polynormer's global attention computation that is $\mathcal{O}(nd^2)$ if $d^2 < n$. However, we find that this is practically not significant for the datasets we considered and hence, together with the above theoretic consideration, can conclude that Principle III is satisfied.

