# OpenReview forum: "Adversarial Robustness of Graph Transformers"
_TMLR — Accepted by TMLR_

### Review · Reviewer_RNcs · 2025-07-10

**Summary Of Contributions:**

In the submitted manuscript, the authors provide a very nice overview of the functioning of five relevant Graph Transformer (GT) architectures. They illustrate how current adversarial attack schemes do not apply to these due to their discrete, non-differentiable nature. The authors then define continuous relaxations of the model equations of the different GTs, which allows them to define both structure perturbation and node injection attacks. The effectiveness of these attacks is proven on several real-world datasets, on which an adversarial attack is somewhat realistic.

**Audience:**

Yes

**Broader Impact Concerns:**

I agree with the broader impact statement that was provided by the authors and have no further concerns to contribute myself.

**Claims And Evidence:**

Yes

**Requested Changes:**

Since my overall impression of this paper is very positive, I only have minor improvement suggestions below. I don't insist on any of these and would be okey with the authors refuting some of them with sufficient arguments.

1] In Figure 1, when you write "The strongest attack for each budget is shown." it would be good to specify out of how many attacks you select the strongest.

2] The following sentence is a bit ambiguous: "Then, by adding the log-probabilities of the edges belonging to one of the attention mechanisms to the attention logits, we obtain: [...]" I think here it would be nice to specify which attention mechanisms are concerned since it was not perfectly clear from the context to me.

3] In Section 4, when you explain your node injection attacks, you write "we connect existing nodes from other graphs of an inductive graph dataset." Here it was not perfectly clear to me whether these nodes are connected to the graphs in isolation (as I suspect) or whether you establish links between graphs in your dataset by maintaining the other connections that the injected node has in its original graph. It may be good to make this explicit.

4] You lack GRIT in the following list: "We investigate the five representative GT models for which we developed continuous relaxations in § 3: Graphormer, SAN, GPS, and Polynormer."

5] I found the repeated word "to" in the following sentence "[...] PRBCD attack to a GCN model to the GT models" confusing.

6] It may be nice to add some further detail here: "Note that for this dataset we were unable to train a comparable Polynormer model." Why were you unable to train this model?

7] Also the "Transferability" setting should be slightly more explicitly defined. The sentence "We collected the adversarial examples generated for each of our adaptive GT attacks and applied them to the other models." was slightly too ambiguous for me to fully grasp which attacks, out of how many attacks, are transferred to what other models. It may be good to explicitly state that all five transformers are considered here.

8] Some references lack detail, e.g., Bamieh (2022) does not have a venue.

**Strengths And Weaknesses:**

Strengths

1] The writing is clear throughout and the work is well-explained. I particularly enjoyed the review of the different Graph Transformers.

2] The continuous relaxations that are applied to the different Graph Transformers are well-motivated and sensible. They also show good empirical performance.

3] I think your selection of datasets (even though you do not have a very large number of them) is sensible.


Weaknesses

1] I struggle to think of any real weakness in this work that needs to be addressed. Overall, the work seems to be clearly described, sensible and proven to work in practice. I suppose one could question whether the presented relaxations are optimal and/or unique. But I do not think this needs to be addressed in the submitted manuscript.

---

> ### Author Response · Authors · 2025-08-02
>
> We want to thank you for the very positive and constructive feedback! We now implemented your requested changes in the updated manuscript. For more details, we refer to our answer below.
>
> #### **R1: Attacks in Figure 1**
>
> We now specify in Figure 1 that the attack is selected out of nine attacks (the adaptive one, the random perturbation and random attack baselines, the GCN transfer baseline, and the transfers from every other GT). We also updated our Section 5 Evaluations to explicitly describe the applied attacks used in Figure 1.
>
> #### **R2: Specification of attention mechanisms**
>
> In the updated paper, we now explicitly state which attention mechanisms are concerned and hope, we could thereby improve our presentation of the relaxed sparse attention mechanism of SAN on Page 6. If there should be any unclarities left, please let us know!
>
> #### **R3: Explicate node injection setting**
>
> As you suspect, we add these nodes in isolation. We now make this explicit in Section 4 in the updated paper.
>
> #### **R4: Missing GRIT in list**
>
> We added GRIT to the list.
>
> #### **R5: Confusing sentence**
>
> We rephrased this sentence in Section 5 in the updated paper for more clarity.
>
> #### **R6: Add details why Polynormer did not train on Reddit Threads**
>
> We now include more details on why we were unable to train Polynormer on Reddit Threads in the Reddit-Threads paragraph on pages 10/11 in the updated manuscript.
>
> In particular, we think that this is related to the fact that the Reddit Threads dataset has no node features. While the other GTs prominently include graph structure information through positional encodings, these are missing in Polynormer. Polynormer only very implicitly considers graph structure through the sparse attention matrix obtained by employing a GAT, which may be insufficient graph inductive bias if node features have no discriminative information.
>
> #### **R7: Define "Transferability" setting more explicitly**
>
> We now explicitly define the "Transferability" setting on Page 11 in the updated paper and also explicitly state that all five transformers are considered. Note that we now also explain the transferability setting in Section 5 Evaluation in the second paragraph associated to "Attacks." (Page 9) to make the attack setting for Figure 1 clear (see first answer).
>
> #### **R8: Some references lack detail**
>
> We have now fixed the references lacking detail in the updated manuscript.
>
> #### **W1: Optimality and/or uniqueness of relaxations**
>
> We think this is an interesting question, raising several challenges. To begin with, it is not clear how to define optimality of a relaxation and defining it w.r.t. the optimal solution to the adversarial objective (Eq. 1 on Page 3) will be NP-hard to measure, even if the objective would be continuous [1]. As a result, we have now added this question as a potential future work direction to our Conclusion on Page 13. The relaxations themselves are not unique. Exemplary, we developed two different ways how to obtain approximate gradients through the eigendecomposition used by the Laplacian PEs in SAN in the case of repeated eigenvalues (see Appendix F.1 & F.2), which we compare in Appendix C.5. Furthermore, we ablated several relaxations for different models in Appendix C.5, where disabling a relaxed component for gradient computation can be understood, as giving rise to a different relaxation of the same model. We want to note that we think that the most crucial point in the context of adversarial attacks against GTs is what characteristics the relaxations have to fulfill, to prove effective in practice - which is the core thought behind our design principles outlined in Section 3.
>
> [1] Katz et al. "Reluplex: An efficient SMT solver for verifying deep neural network", CAV 2017

---

> > ### Comment · Reviewer_RNcs · 2025-08-12
> >
> > I want to thank the authors for their rebuttal. Their responses comprehensively address my comments.

---

> > > ### Author Response · Authors · 2025-08-18
> > >
> > > Thank you for your response and for noting that we comprehensively addressed your comments!

---

### Review · Reviewer_SPiU · 2025-07-20

**Summary Of Contributions:**

This paper investigates the adversarial robustness of Graph Transformers (GTs), proposing a gradient-based adaptive attacks specifically tailored for GT architectures. It analyzes vulnerabilities across various GT architectures featuring different attention mechanisms and Positional Encodings (PEs), demonstrating significant fragility to adversarial perturbations. The authors introduce general principles to facilitate continuous relaxations, which are necessary for these gradient-based attacks. Furthermore, they leverage these adaptive attacks to improve GT robustness substantially. The empirical evaluation across diverse datasets confirms both the efficacy of adaptive attacks and the superiority of the adversarial training approach.

**Audience:**

Yes

**Broader Impact Concerns:**

There are no broader impact concerns.

**Claims And Evidence:**

Yes

**Requested Changes:**

Principle II is not obvious. The observation below the principle states "We argue that in Principle II, we do not require continuous differentiability, as to obtain informative gradients w.r.t. the input data, we do not need to enforce stronger standards on ˜fθ than perhaps the most widely used activation function." Can authors further elaborate on why the requirement is only for the function to be "differentiable almost everywhere" and not "differentiable" (or for that matter only continuous with no differentiability requirement at all).

For equation 17, "Note that for a discrete connected edge A˜′ij = 1, the log-probabilities become 0, or −∞ respectively. Thus, such an edge still fully contributes to the connected attention mechanism (over Ni), while not affecting the disconnected one (over V \ Ni)" - doesn't this lead to numerical instability? Is the choice of log function appropriate in Eqn 17? Wouldn't a function that saturates be better?

**Strengths And Weaknesses:**

+ Proposes novel adaptive gradient-based attacks specifically tailored to Graph Transformers, effectively addressing gaps left by prior work primarily focused on MPNNs

+ Provides insights into the continuous relaxations required for effective gradient-based robust attacks, with a focus on GT architectures.

+ Demonstrates significant robustness improvements through adversarial training validated by extensive empirical evaluation against baseline approaches.

- The focus is rather narrow - untargeted global evasion attacks, which makes the practical applicability of the work very narrow. It does reduce the need for comparison with other interesting baselines even though the core method presented could have had a broader attack relevance and ideally have a more extensive baseline comparison (e.g. Netattack - [6] D. Zügner, A. Akbarnejad, S. Günnemann, "Adversarial Attacks on Neural Networks for Graph Data", KDD 2018).

- The theoretical part of the paper is not well-explained and some questions below would help improve the presentation. It is not essential for the paper to have a principled/theoretically justified approach if the empirical results are strong and the paper has practical relevance.

---

> ### Author Response · Authors · 2025-08-02
>
> We want to thank you for your helpful feedback! We now include the requested changes in the updated draft and address all requested changes and weaknesses in detail below.
>
> ### W1: Narrow practical applicability due to focus on untargeted global evasion attacks
>
> We respectfully disagree that our work only has narrow practical applicability, in particular we want to highlight that:
>
> **1)** We do not only focus on untargeted global attacks, which are applied to node-classification tasks, but we also focus on node-injection attacks for graph classification, as well as using our attacks for effective adversarial training.
>
> **2)** Untargeted global evasion attacks are the most general graph modification attacks for node-classification tasks and the main focus of many high-impact graph robustness works [1,2,3,4]. In particular, as you also mentioned, any global attack can always be used to attack a single node by restricting the set of nodes attacked to the target node, i.e., it can be used as a local attack like Nettack. Thus, in the updated manuscript, we now also employ our global attacks as local attacks, and report the results in Appendix C.7 (Page 25) and reference them in the main paper on Page 10. The results highlight the efficacy of our adaptive attacks in a local attack setting. Additionally, we are currently working on implemented Nettack and will update the draft with this baseline comparison, once we obtain the results.
>
> **3)** The key contribution of our work is how to enable effective differentiation through graph transformers w.r.t. the discrete adjacency matrix for which we provide general guiding principles,  and which we instantiate for five representative GT architectures. We then choose the very effective tanh-margin loss from [1] as an untargeted attack loss, which for a particular node reads $\tanh(\max\_{c \ne c^\*} z\_c - z\_{c^\*})$, where $z\_c$ refers to the logit of class $c$ outputed by the GT for that node and $c^\*$ to the original class. However, as our contribution now allows to differentiate through GTs w.r.t. the input data, one is free to choose any other attack-loss function to optimize, including targeted attack loss functions such as $\tanh(z\_t - z\_{c^\*})$ or any other targeted loss function [5], where $t$ is the target class. Thus, *our method also enables adaptive targeted evasion attacks*.
>
> [1] Xu et al. "Topology attack and defense for graph neural networks: An optimization perspective", IJCAI 2019.
> [2] Ma et al. "Towards More Practical Adversarial Attacks on Graph Neural Networks", NeurIPS 2020.
> [3] Geisler et al. "Robustness of Graph Neural Networks at Scale", NeurIPS 2021.
> [4] Gosch et al. "Adversarial training for graph neural networks: Pitfalls, solutions, and new directions", NeurIPS 2023.
> [5] Carlini et al. "Towards Evaluating the Robustness of Neural Networks", IEEE S&P 2017.

---

> > ### Author Response · Authors · 2025-08-02
> >
> > ### R1: Elaborate on differentiability in Principle II
> >
> > We now expanded our discussion on the differentiability requirement in Principle II in the main draft (see Page 5, Section 3) and also added an Appendix G.1 (Pages 30-31), where we outline in detail why we require "differentiable almost everywhere" and also not only continuity.
> >
> > In essence, continuous differentiability is an unnecesarry strict requirement to effectively optimize a loss function with stochastic gradient descent. In particular, having a $(i)$ continuous, and $(ii)$ almost everywhere differentiable function sufficies. This can be best understood using the ReLU function as an analogy. It satisfies both $(i)$ and $(ii)$, but is not continuously differentiable - still, neural networks employing ReLU can be effectively optimized (i.e., trained) using gradient descent. This is possible as for any function satisfying $(i)$ and $(ii)$, the gradient can be computed for all inputs except on a subset having measure zero, which for ReLU is $\{0\}$. In practice, the non-differentiable points are nearly never reached, and if so, due to the continuity, a "gradient" can still be defined at every non-differentiable point $x_0$ in a principled way, by choosing it to be either the left-sided limit $a=\lim_{x\rightarrow x_0^-} \frac{f(x)-f(x_0)}{x-x_0}$ or the right-sided limit $b=\lim_{x\rightarrow x_0^+} \frac{f(x)-f(x_0)}{x-x_0}$ of the function. Thus, the non-differentiability on a measure zero set is not interfering with effective continuous optimization through methods like gradient descent. For ReLU, the left-sided limit is $0$, which e.g., the current PyTorch version returns if you want to take the gradient of ReLU at $0$.
> >
> > We do require differentiability almost everywhere and not only continuity, as otherwise, gradient-free optimization methods have to be used to solve the attack objective (Eq. 1 on page 3 in the paper), which is a complex and NP-hard combinatorial optimization problem. Indeed, the literature on adversarial attacks for GNNs has shown that the most effective attacks against GNNs are usually gradient-based attacks [1,2,3], which is consistent with the image domain [4].
> >
> > [1] Xu et al. "Topology attack and defense for graph neural networks: An optimization perspective", IJCAI 2019
> > [2] Geisler et al. "Robustness of graph neural networks at scale", NeurIPS 2021.
> > [3] Gosch et al. "Adversarial training for graph neural networks: Pitfalls, solutions, and new directions", NeurIPS 2023.
> > [4] Tramer et al. "On adaptive attacks to adversarial example defense", NeurIPS 2020.

---

> > > ### Author Response · Authors · 2025-08-02
> > >
> > > ### R2: Numerical instability and choice of log function
> > >
> > > Thank you for this question! Given proper treatment, it does not lead to numerical instabilites, but before going into the details of the computation, we want to note that the choice of the log-function is motivated by the fact that in the relaxed model, we have continous edge weights $\tilde{A}\_{ij} \in [0,1]$, which correspond to the probability $p\_{ij}=\tilde{A}\_{ij}$ that nodes $i$ and $j$ are connected when we sample a final perturbation matrix. Thus, we want that the local attention score $\alpha\_{ij}$ computed by SAN (or other GNNs and GTs) reflects this *probability* of the edge being in the final graph. Usually, an edge $(i,j)$ with attention logit $w\_{ij}$ contributes $e^{w\_{ij}}$ to the attention score calculation. Thus, to include the probabilty of being connected, we scale $e^{w\_{ij}}$ by $p\_{ij}$, i.e., the edge now contributes $p\_{ij}e^{w\_{ij}}$ to the attention score calculation, which then changes to $\tilde{\alpha}\_{ij}=\frac{p\_{ij}e^{w\_{ij}}}{\sum\_k p\_{ik} e^{w\_{ik}}}$. As a result, if the edge is known to be in the graph, i.e., $p\_{ij}=1$, the edge fully contributes to the attention score as in the original model and if the edge is not in the graph, i.e. $p\_{ij}=0$, the node $j$ is not attended to ($\tilde{\alpha}\_{ij}=0$), again as in the original model (Principle I). For in-between values $p\_{ij}\in(0,1)$, the above attention score formulation represents a well-behaved interpolation (Principle II) between the two binary outcomes of the sparse attention:
> > >
> > > $$ \alpha\_{ij} =\frac{e^{w\_{ij}}}{\sum\_{k \in \mathcal{N}\_i} e^{w\_{ik}}} \text{ if } A\_{ij}=1 \ \land \
> > > 0\ \text{otherwise.}$$
> > >
> > > Regarding the computation aspect, the logarithm now comes into play, because one can rewrite the relaxed attention score calculation as $\tilde{\alpha}\_{ij}=\frac{p\_{ij}e^{w\_{ij}}}{\sum\_k p\_{ik} e^{w\_{ik}}} = \frac{e^{w\_{ij}+\log p\_{ij}}}{\sum\_k e^{w\_{ik}+\log p\_{ik}}} = \frac{e^{\tilde{w}\_{ij}}}{\sum\_k e^{\tilde{w}\_{ik}}} = softmax(\tilde{w}\_{ij})$ with $\tilde{w}\_{ij}=w_{ij} + \log p\_{ij}$. Then, this can also be understood as a masked attention calculation [1] if $p\_{ij}=0$. Strictly speaking, this is not necessary and one could directly work with $\tilde{\alpha}\_{ij}=\frac{p\_{ij}e^{w\_{ij}}}{\sum\_k p\_{ik} e^{w\_{ik}}}$. In our implementation, we work with the $\log$-formulation and directly use PyTorch's softmax implementation. It knows how to deal with $-\infty$ as an input [2] as is common in masked attention, and its calculation is numerically stable. However, to ensure that the gradient-taking w.r.t. $p\_{ij}$ of our "soft mask" $\log p\_{ij}$ is also numerically well-behaved, we set the minimum value of $p\_{ij}$ to some small $\epsilon$ (we choose $10^{-9}$) and hence, $p\_{ij}=0$ implying $\tilde{w}\_{ij}=-\infty$ is never reached. This follows the strategy employed by the attack framework PR-BCD [3], which we leverage to execute our attacks (see Section 5 in the paper). For scalability, PR-BCD iteratively samples a batch of edges and only computes the gradient w.r.t. these edges. It always sets the minimum value of a sampled edge to some small $\epsilon$, to ensure proper gradients can be calculated, even if the edge does not originally exist in the graph.
> > >
> > > **Action:** Based on your question, we have now expanded the motivation behind choosing the $\log$-function in the paper, as well as included a discussion on the numerical stability in our description of the relaxed attention mechanism on Page 6 and a newly introduced Appendix G.2 (Page 31) in the updated manuscript.
> > >
> > > Thus, to conclude, adding $\log p\_{ij}$ as a bias in the attention weight calculation is conceptually well-motivated, works well in practice (see our attack results in Section 6 of the paper), and we do not observe any significant numerical issues. We hope the new additions to the paper make this now clear to the reader, and if you think that other relaxations or function choices would be interesting, we are happy to hear concrete suggestions.
> > >
> > > [1] Cheng et al. "Masked-attention Mask Transformer for Universal Image Segmentation", CVPR 2022.
> > > [2] https://docs.pytorch.org/docs/stable/generated/torch.nn.Softmax.html.
> > > [3] Geisler et al. "Robustness of graph neural networks at scale", NeurIPS 2021.

---

> > > > ### Author Response · Authors · 2025-08-02
> > > >
> > > > ### W2 & R1-2: Presentation of the theoretical part
> > > >
> > > > We took great effort to make our paper and methods as easy to follow and as well-explained as possible. We hope that by addressing your requested changes, we could further improve our presentation and the clarity of our paper. If there are other presentation issues you feel we should address, we are happy to do so!
> > > >
> > > > ### Checkbox: Claims And Evidence
> > > >
> > > > We noticed that you put a "No" at "Claims And Evidence". We wondered if, after our rebuttal, any claims are remaining for which you feel that we do not provide appropriate evidence. We would very much appreciate and be happy to get aware of such claims so that we can adapt them in the paper, as we tried to carefully formulate our claims in accordance with our contributions and results we obtained.

---

> > > > > ### Comment · Reviewer_SPiU · 2025-08-17
> > > > > **Thank you**
> > > > >
> > > > > My concerns have been met and I have raised the score to accept.

---

> > > > > > ### Author Response · Authors · 2025-08-18
> > > > > >
> > > > > > Thank you for your response and for raising the score to accept!

---

> > > > > ### Author Response · Authors · 2025-08-17
> > > > > **Update on W1**
> > > > >
> > > > > We want to make you aware that the now updated draft also includes Nettack as a baseline in the local attack setting (Appendix C.7, Page 25). If there should be any open questions left, we are happy to engage in further discussion!

---

### Review · Reviewer_q7D1 · 2025-07-22

**Summary Of Contributions:**

This work studies the adversarial robustness of graph transformers (GT). The authors design general principles and differentiable relaxations for adaptive gradient-based structure attacks tailored to GTs, accounting for their unique attention mechanisms and positional encodings. They instantiate these attacks for five representative GT architectures and evaluate them on multiple node and graph-level tasks, including structure perturbations and node injection attacks. The results reveal that GTs can be catastrophically vulnerable to adversarial attacks—sometimes even more so than classic GNNs. However, the study also demonstrates that adversarial training, when done with these adaptive attacks, can significantly improve the robustness of GTs, often surpassing traditional GNNs.

**Audience:**

Yes

**Claims And Evidence:**

Yes

**Requested Changes:**

1. It's unclear whether the generated perturbations are semantically unnoticeable or not. It'd be great if certain proof-of-concept evidence could be provided, as it's been shown that the adversarial attack without constraint may destroy the original graph distribution by [1]

[1] Understanding and Improving Graph Injection Attack by Promoting Unnoticeability, ICLR'22.

2. The attack is limited to white-box setting, which may not be practical. It's intereseting to also study the transferability across different GTs, in addition to the comparison between GTs and message-passing based GNNs.

3. The evaluation with respect to the improvements in robustness of adversarial training may not be fair. The authors claim that "he flexibility of GT models can lead to significantly better robust learning capabilities compared to classic message-passing GNNs.", while it seems message-passing GNNs are not trained with adversarial attacks.

4. The proposed attack is also limited in terms of the sizes of the graphs, due to its limitations in efficiency and scalability.

**Strengths And Weaknesses:**

## Strengths

(+) This work provides and instantiates the first comprehensive empirical studies on the adversarial robustness of GTs;

(+) The authors develop several principles and relaxations in order to perform the adversarial attacks on GTs;

(+) Empirical results demonstrate the effectiveness of the proposed attack methods;

## Weakness

(-) It's unclear whether the generated perturbations are semantically unnoticeable or not;

(-) The attack is limited to white-box setting, which may not be practical;

(-) The evaluation with respect to the improvements in robustness of adversarial training may not be fair;

---

> ### Author Response · Authors · 2025-08-02
>
> We want to thank you for the helpful and constructive feedback! Below, we address your comments and requested changes, which we also included in our updated draft. For more details, we refer to our answers below.
>
> #### R1 & W1: Semantic unnoticability
>
> Thank you for this question! We now provide the requested evidence that shows that our attacks are semantically unnoticeable in the updated manuscript. In particular, we now provide the node-centric homophily measurements used by [1] to measure unnoticability, for our attacks for the UPFD gossipcop and politifact dataset in the newly added Appendix C.6 (Pages 23/24) and refer to it in our results section on Page 10.
>
> **Explanation:** Concretely, [1] highlights that unnoticability can be a problem for node-injection attacks. Our measurements show that our node-injection attacks against all graph transformers are semantically unnoticeable by the criterion defined in [1] and thus, their "trivial" defense of cutting away all nodes with a node-centric homophily less than that found in the training set won't be effective against our attacks. This can be understood by the fact that while [1] also deals with node-insertion attacks, they allow the node features to be optimized over arbitrarily in the support of the original data. However, we do not optimize over the features and only insert nodes that already exist in other graphs of the same dataset.
>
> **Example:** For the UPFD dataset, nodes correspond to real users, and inserting a node into a graph through our attack represents another retweet by a potentially adversarially controlled user, but with realistic node-features instead of arbitrarily setting them. Concerning node-centric homophily, this is more related to graph modification attacks, for which [1] show that the noticability problem is not existing. Thus, this makes for a semantically unnoticeable perturbation. For more details on our unnoticability results, we refer to Appendix C.6.
>
> [1] Chen et al. "Understanding and Improving Graph Injection Attack by Promoting Unnoticeability", ICLR 2022
>
> #### R2 & W2: White-box setting and transferability across different GTs
>
> The requested detailed transferability study across different GTs can be found in Appendix C.4  (Pages 20-21). It highlights that depending on the dataset, some GT models (e.g., Graphormer and GRIT, see Fig. 15d & 16d) transfer well, while others do not (e.g., GPS to Graphormer, see Fig. 15d). While the study was already included in the original submission, we now improved the reference to these results on Page 11 in our results section, to highlight these results more explicitly in the main draft.
>
> Note that the goal of our work is to understand the worst-case robustness of existing GT models and provide general guidelines on how to design and perform such an investigation for GT modules in a principled manner, following the best practices of the field [1]. For this, the white-box setting as the strongest threat model is the most appropriate, as otherwise, vulnerabilities could be overlooked [2,3]. Additionally, strong adaptive attacks as developed for a principled adversarial study in a white-box setting, as done in our work, are the foundation to perform effective adversarial training [1], the effectiveness of which we show in Section 7 Adversarial Training. Thus, the white-box setting indeed is *practical from the perspective of the defender*.
>
> [1] Tramer et al. "On Adaptive Attacks to Adversarial Example Defenses", NeurIPS 2020
> [2] Carlini and Wagner, "Adversarial Examples Are Not Easily Detected: Bypassing Ten Detection Methods", AISec 2017
> [3] Mujkanovic et al. "Are Defenses for Graph Neural Networks Robust?", NeurIPS 2022
>
> #### R3 & W3: Message-passing GNNs are not trained with adversarial attacks.
>
> We are sorry for this misunderstanding, in fact **we did train the compared-to message-passing GNNs with adversarial attacks** to have fair comparisons. In particular, in Figure 7 (Page 12) in the paper, we compare Graphormer normally and adversarially trained with a GCN normally and adversarially trained. Specifically, Figures 7a and 7c show the results of training a GCN normally and with two (Figure 7a) and three (Figure 7c) different adversarial training configurations, the same as used for Graphormer. Thus, the evaluation and comparison are fair.
>
> Based on your comments, we have now added some clarifications to Section 7 Adversarial Training and to the description of Figure 7, to make it more explicit that we indeed train the GCN we compare to adversarially.

---

> > ### Author Response · Authors · 2025-08-02
> >
> > ### R4: Scalability
> >
> > Our attacks do not introduce any particular inefficiencies or limitations on scalability that are not already inherent to the graph transformer models. In particular, except for the Polynormer model, all studied graph transformers have at least quadratic runtime w.r.t. the number of nodes in the graph (due their global full-node attention), which is *the major limitation* in terms of the efficiency and scalability of GT models, and hence, also of our attacks, to larger graphs. Specifically, our relaxations do not change the asymptotic runtime complexities of these models and mostly only add minor overhead calculations, which are dominated by the general GT computations done even without our relaxations. Thus, given more scalable GTs - which is an active research area [1], developing relaxations following our design principles will automatically yield scalable attacks.
> >
> > To make this more clear in the paper, we have now added a paragraph on "Computational Complexity" on page 8 to Section 3 Attacking Graph Transformers. There, we further refer to the newly added Appendix G.3 (Pages 31-32), where we discuss in detail the computational effects and runtime complexities of our introduced relaxations.
> >
> > One should note that when attacking GNNs, gradient-based attacks usually take several iterations of forward and backward passes and thus, evaluate a model multiple times. This is inherent to the gradient-based attacks [2] and not to our method in particular. This is also the reason for Principle III, where we state that any relaxed model must be efficient and not excessively increase memory or runtime complexity, which we show fulfilled for our relaxations in Appendix G.3.
> >
> > [1] Müller et al. "Attending to Graph Transformers", TMLR 2024.
> > [2] Geisler et al. "Robustness of Graph Neural Networks at Scale", NeurIPS 2021

---

> > > ### Comment · Reviewer_q7D1 · 2025-08-05
> > >
> > > Thanks the authors for the detailed explanations. They resolved my concerns well.

---

> > > > ### Author Response · Authors · 2025-08-18
> > > >
> > > > Thank you for your response! We are happy to have resolved your concerns!

---

### Author Response · Authors · 2025-08-02
**General Comment**

We want to thank the reviewers for their helpful and constructive feedback, highlighting the comprehensiveness of our study and effectiveness of our approach (Reviewer q7DI), the effectively addressed gap left by prior work (Reviewer SPiU), as well as the clear writing and well-motivated relaxations (Reviewer RNcs). Based on the feedback, we now include several new interesting results and discussions in the paper addressing the requested changes and weaknesses. We highlight updates to the paper in blue.

Concretely, we added the following **new results**:
* We measure and show the **semantic unnoticability** [1] of our attacks (Appendix C.6).
* We apply our adaptive attacks in a **local attack setting** (Appendix C.7)

Additonally, we did the following **updates to the draft**:
* Highlight the transferability study across different GTs (Appendix C.4) in Section 6.
* Added a discussion on the computational complexity of our relaxations (Appendix G.3).
* Added a detailed discussion on the differentiability and continuity aspects of Principle II (Appendix G.1)
* Improved the presentation of the relaxed sparse attention of SAN, including a discussion of the numerical stability in Appendix G.2
* Added several minor clarifications and changes in the main draft, including the adversarial training of message-passing GNNs, the experimental setting of Fig. 1, and others.

We are positive to have addressed the requested changes and comments by the reviewers. If any open questions or concerns remain, we are happy to engage in further discussion.

[1] Chen et al. "Understanding and Improving Graph Injection Attack by Promoting Unnoticeability", ICLR 2022

---

> ### Author Response · Authors · 2025-08-17
>
> *Update 17.8.:* We now also include Nettack as a baseline in the local attack setting (Appendix C.7)

---

### Decision · Action_Editor_YPNp · 2025-08-29

**Recommendation:** Accept as is

**Additional Comments:**

The paper presents the first comprehensive study on the adversarial robustness of Graph Transformers (GTs). The authors design a framework for creating adaptive, gradient-based adversarial attacks specifically tailored to five popular GT architectures. Reviewers acknowledge the novelty and impact as the first work to systematically investigate the adversarial robustness of graph transformers (I think it's worth mentioning even though novelty and impact are not TMLR criteria) and the extensivenss of empirical analysis, and most initial concerns have been addressed during rebuttal and in the revision and I am happy to recommend acceptance as-is. My only, optional suggestion to the authors is that some reviewers' remarks like the ones about attack practicality and scope are important, and thus I suggest the authors to consider moving some of the discussions to the main text but I leave the ultimate judgement to the authors.

**Audience:**

Yes

**Audience Explanation:**

All reviewers agreed that the submission is of interest to TMLR audience. The AE agrees that this paper clearly meets this criterion.

**Claims And Evidence:**

Yes

**Claims Explanation:**

The reviewers acknowledged that the authors conducted extensive experiments on multiple tasks and datasets, providing strong empirical evidence for their claims. There were some initial concerns from reviewers about the justification of "differentiable almost everywhere" requirement (SPIU), concerns about fairness of comparisons to GNNs and whether the attack preserved the graph's semantic distribution (q7D1) -- these concerns have been addressed to a satisfactory extent during rebuttal and all reviewers appreciate the soundness of the paper at the end of the review process.